# Discovering Invariant Rationales for Graph Neural Networks

**Ying-Xin Wu**[†], **Xiang Wang**[*†], **An Zhang**[§], **Xiangnan He**[†], **Tat-Seng Chua**[§]

[†] University of Science and Technology of China

[§] National University of Singapore

{wuyxinsh, xiangwang1223}@gmail.com,
an_zhang@nus.edu.sg, xiangnanhe@gmail.com, dcscts@nus.edu.sg

## Abstract

Intrinsic interpretability of graph neural networks (GNNs) is to find a small subset of the input graph's features — rationale — which guides the model prediction. Unfortunately, the leading rationalization models often rely on data biases, especially shortcut features, to compose rationales and make predictions without probing the critical and causal patterns. Moreover, such data biases easily change outside the training distribution. As a result, these models suffer from a huge drop in interpretability and predictive performance on out-of-distribution data. In this work, we propose a new strategy of discovering invariant rationale (DIR) to construct intrinsically interpretable GNNs. It conducts interventions on the training distribution to create multiple interventional distributions. Then it approaches the causal rationales that are invariant across different distributions while filtering out the spurious patterns that are unstable. Experiments on both synthetic and real-world datasets validate the superiority of our DIR in terms of interpretability and generalization ability on graph classification over the leading baselines. Code and datasets are available at https://github.com/Wuyxin/DIR-GNN.

## 1 Introduction

The eye-catching success in graph neural networks (GNNs) (Hamilton et al., 2017; Kipf & Welling, 2017; Dwivedi et al., 2020) provokes the rationalization task, answering "What knowledge drives the model to make certain predictions?". The goal of selective rationalization (*aka.* feature attribution) (Chang et al., 2020; Ying et al., 2019; Luo et al., 2020; Wang et al., 2021c) is to find a small subset of the input's graph features — *rationale* — which best guides or explains the model prediction. Discovering the rationale in a model helps audit its inner workings and justify its predictions. Moreover, it has tremendous impacts on real-world applications, such as finding functional groups to shed light on protein structure prediction (Senior et al., 2020).

Two research lines of rationalization have recently emerged in GNNs. Post-hoc explainability (Ying et al., 2019; Luo et al., 2020; Yuan et al., 2021; Wang et al., 2021c) attributes a model's prediction to the input graph with a separate explanation method, while intrinsic interpretability (Veličković et al., 2018; Gao & Ji, 2019) incorporates a rationalization module into the model to make transparent predictions. Here we focus on intrinsically interpretable GNNs. Among them, graph attention (Veličković et al., 2018) and pooling (Lee et al., 2019; Knyazev et al., 2019; Gao & Ji, 2019; Ranjan et al., 2020) operators prevail, which work as a computational block of a GNN to generate soft or hard masks on the input graph. They cast the learning paradigm of GNN as minimizing the empirical risk with the masked subgraphs, which are regarded as rationales to guide the model predictions.

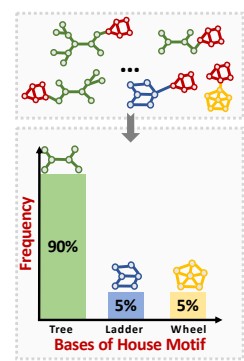

Figure 1: Base Distribution of *House* Motif.

Despite the appealing nature, recent studies (Chang et al., 2020; Knyazev et al., 2019) show that the current rationalization methods are prone to

---

*Corresponding author.

exploit data biases as shortcuts to make predictions and compose rationales. Typically, shortcuts result from confounding factors, sampling biases, and artifacts in the training data. Considering Figure 1, when the most bases of *House*-motif graphs are *Tree*, a GNN does not need to learn the correct function to reach high accuracy for the motif type. Instead, it is much easier to learn from the statistical shortcuts linking the bases *Tree* with the most occurring motifs *House*. Unfortunately, when facing with out-of-distribution (OOD) data, such methods generalize poorly since the shortcuts are changed. Hence, such shortcut-involved rationales hardly reveal the truly critical subgraphs for the predicted labels, being at odds with the true reasoning process that underlies the task of interest (Teney et al., 2020) and human cognition (Alvarez-Melis & Jaakkola, 2017).

Here we ascribe the failure on OOD data to the inability to identify causal patterns, which are stable to distribution shift. Motivated by recent studies on invariant learning (IL) (Arjovsky et al., 2019; Krueger et al., 2021; Chang et al., 2020; Bühlmann, 2018), we premise different distributions elicit different environments of data generating process. We argue that the causal patterns to the labels remain stable across environments, while the relations between the shortcut patterns and the labels vary. Such environment-invariant patterns are more plausible and qualified as rationales.

Aiming to identify rationales that capture the environment-invariant causal patterns, we formalize a learning strategy, Discovering Invariant Rationales (DIR), for intrinsically interpretable GNNs. One major problem is how to get multiple environments from a standard training set. Differing from the heterogeneous setting (Bühlmann, 2018) of existing IL methods, where environments are observable and attainable, DIR does not assume prophets about environments. It instead generates distribution perturbations by causal intervention — interventional distributions (Tian et al., 2006; Pearl et al., 2016) — to instantiate environments and further distinguish the causal and non-causal parts.

Guided by this idea, our DIR strategy consists of four modules: a rationale generator, a distribution intervener, a feature encoder, two classifiers. Specifically, the rationale generator learns to split the input graph into causal and non-causal subgraphs, which are respectively encoded by the encoder into representations. Then, the distribution intervener conducts the causal interventions on the non-causal representations to create perturbed distributions, with which we can infer the invariant causal parts. Then, the two classifiers are respectively built upon the causal and non-causal parts to generate the joint prediction, whose invariant risk is minimized across different distributions. On one synthetic and three real datasets, extensive experiments demonstrate the generalization ability of DIR to surpass current state-of-the-art IL methods (Arjovsky et al., 2019; Krueger et al., 2021; Sagawa et al., 2019), and the interpretability of DIR to outperform the attention- and pooling-based rationalization methods (Veličković et al., 2018; Gao & Ji, 2019). Our main contributions are:

- We propose a novel invariant learning algorithm, DIR, for inherent interpretable models, improving the generalization ability and is suitable for any deep models.
- We offer causality theoretic analysis to guarantee the preeminence of DIR.
- We provide the implementation of DIR for graph classification tasks, which consistently achieves excellent performance on three datasets with various generalization types.

## 2 INVARIANT RATIONALE DISCOVERY

With a causal look at the data-generating process, we formalize the principle of discovering invariant rationales, which guides our discovery strategy. Throughout the paper, upper-cased letters like $G$ denote random variables, while lower-case letters like $g$ denote deterministic value of variables.

### 2.1 CAUSAL VIEW OF DATA-GENERATING PROCESS

Generating rationales for transparent predictions requires understanding the actual mechanisms of the task of interest. Without loss of generality, we focus on the graph classification task and present a causal view of the data-generating process behind this task. Here we formalize the causal view as a Structure Causal Model (SCM) (Pearl et al., 2016; Pearl, 2000) by inspecting on the causalities among four variables: input graph $G$, ground-truth label $Y$, causal part $C$, non-causal part $S$. Figure 2a illustrates the SCM, where each link denotes a causal relationship between two variables.

- $C \rightarrow G \leftarrow S$. The input graph $G$ consists of two disjoint parts: the causal part $C$ and the non-causal part $S$, such as the *House* motif and the *Tree* base in Figure 1.

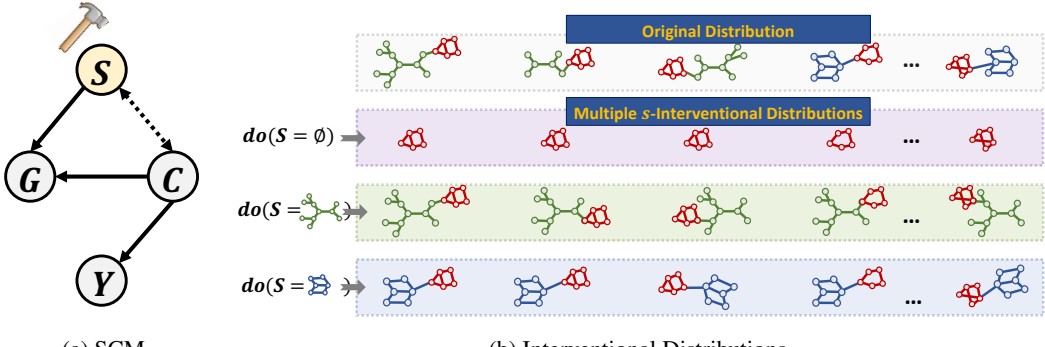

(a) SCM                (b) Interventional Distributions.

Figure 2: (a) Causal view of data-generating process; (b) Illustration of interventional distributions.

- $C \to Y$. By "causal part", we mean $C$ is the only endogenous parent to determine the ground-truth label $Y$. Taking the motif-base example in Figure 1 again, $C$ is the oracle rationale, which perfectly explains why the graph is labeled as $Y$.
- $C \dashleftarrow\dashrightarrow S$. This dashed arrow indicates additional probabilistic dependencies (Pearl, 2000; Pearl et al., 2016) between $C$ and $S$. We consider three typical relationships here: (1) $C$ is independent of $S$, *i.e.*, $C \perp\!\!\!\perp S$; (2) $C$ is the direct cause of $S$, *i.e.*, $C \to S$; and (3) There exists a common cause $E$, *i.e.*, $C \leftarrow E \to S$. See Appendix B for the corresponding examples.

$C \dashleftarrow\dashrightarrow S$ can create spurious correlations between the non-causal part $S$ and the ground-truth label $Y$. Assuming $C \to S$, $C$ is a confounder between $S$ and $Y$, which opens a backdoor path $S \leftarrow C \to Y$, thus making $S$ and $Y$ spuriously correlated (Pearl et al., 2016). We systematize such spurious correlations as $Y \not\perp\!\!\!\perp S$. Wherein, we make feature induction assumption on $S$ to avoid the confusion of the induced subset of $S$ between $C$. See Appendix C for the formal assumption. Furthermore, data collected from different environments exhibit various spurious correlations (Teney et al., 2020; Arjovsky et al., 2019), *e.g.,* one mostly picks *House* motifs with *Tree* bases as the training data, while another selects *House* motifs with *Wheel* bases as the testing data. Hence, such spurious correlations are unstable and variant across different distributions.

## 2.2 TASK FORMALIZATION OF INVARIANT RATIONALIZATION

**Oracle Rationale.** With the causal theory (Pearl et al., 2016; Pearl, 2000), for each variable $X$ in a SCM, there exists a directed link from each of its parent variables $PA(X)$ to $X$, if and only if the causal mechanism $X = f_X(PA(X), \epsilon_X)$ persists, where $\epsilon_X \perp\!\!\!\perp PA(X)$ is the exogenous noise of $X$. For simplicity, we omit the exogenous noise and simplify it as $X = f_X(PA(X))$. Hence, there exist a function $f_Y : C \to Y$ in our SCM, where the "oracle rationale" $C$ satisfies:

$$Y = f_Y(C), \quad Y \perp\!\!\!\perp S \mid C, \tag{1}$$

where $Y \perp\!\!\!\perp S \mid C$ indicates that $C$ shields $Y$ from the influence of $S$, making the causal relationship $C \to Y$ invariant across different $S$.

**Rationalization.** In general, only the pairs of input $G$ and label $Y$ are observed during training, while neither oracle rationale $C$ nor oracle structural equation model $f_Y$ is available. The absence of oracles calls for the study on intrinsic interpretability. We systematize an intrinsically-interpretable GNN as a combination of two modules, *i.e.*, $h = h_{\hat{Y}} \circ h_{\tilde{C}}$ , where $h_{\tilde{C}} : G \to \tilde{C}$ discovers rationale $\tilde{C}$ from the observed $G$, and $h_{\hat{Y}} : \tilde{C} \to \hat{Y}$ outputs the prediction $\hat{Y}$ to approach $Y$. Distinct from $C$ and $Y$ which are the variables in the causal mechanisms, $\tilde{C}$ and $\hat{Y}$ represent the variables in the modeling process to approximate $C$ and $Y$. To optimize these modules, most of current intrinsically-interpretable GNNs (Veličković et al., 2018; Lee et al., 2019; Knyazev et al., 2019; Gao & Ji, 2019; Ranjan et al., 2020) adopt the learning strategy of minimizing the empirical risk:

$$\min_{h_{\tilde{C}}, h_{\hat{Y}}} \mathcal{R}(h_{\hat{Y}} \circ h_{\tilde{C}}(G), Y), \tag{2}$$

where $\mathcal{R}(\cdot, \cdot)$ is the risk function, which can be the cross-entropy loss. Nevertheless, this learning strategy relies heavily on the statistical associations between the input features and labels, and can potentially exhibit non-causal rationales.

**Invariant Rationalization.** We ascribe the limitation to ignoring $Y \perp\!\!\!\perp S \mid C$ in Equation 1, which is crucial to refine the causal relationship $C \rightarrow Y$ that is invariant across different $S$. By introducing this independence, we formalize the task of invariant rationalization as:

$$\min_{h_{\tilde{C}}, h_{\hat{Y}}} \mathcal{R}(h_{\hat{Y}} \circ h_{\tilde{C}}(G), Y), \quad \text{s.t. } Y \perp\!\!\!\perp \tilde{S} \mid \tilde{C}, \tag{3}$$

where $\tilde{S} = G \setminus \tilde{C}$ is the complement of $\tilde{C}$. This formulation encourages the rationale $\tilde{C}$ seeking the patterns that are stable across different distributions, while discarding the unstable patterns.

## 2.3 PRINCIPLE & LEARNING STRATEGY OF DIR

**Interventional Distribution.** However, it is difficult to recover the oracle rationale from the joint distribution over the inputs and labels — that is, the causal and non-causal relations are hardly distinguished from each other. We get inspirations from invariant learning (Arjovsky et al., 2019; Krueger et al., 2021; Chang et al., 2020), which constructs different environments to infer the invariant features or predictors. To obtain the environments, previous studies mostly partition the training set by prior knowledge (Teney et al., 2020) or adversarial environment inference (Creager et al., 2021; Wang et al., 2021b). Different from partitioning the training data, we do not assume prophets about environments but introduce the interventional distribution (Tian et al., 2006; Pearl et al., 2016) instead to model the DIR task. Specifically, on the top of our SCM, we generate $s$-interventional distribution by doing intervention $do(S = s)$ on $S$, which removes every link from the parents $PA(S)$ to the variable $S$ and fixes $S$ to the specific value $s$. By stratifying different values $\mathbb{S} = \{s\}$, we can obtain multiple $s$-interventional distributions.

With interventional distributions, we propose the principle of discovering invariant rationale (DIR) to identify a rationale $\tilde{C}$ whose relationship with the label $Y$ is stable across different distributions.

**Definition 1 (DIR Principle)** *An intrinsically-interpretable model $h$ satisfies the DIR principle if it*

    *1. minimizes all $s$-interventional risks: $\mathbb{E}_s[\mathcal{R}(h(G), Y|do(S = s))]$, and simultaneously*

    *2. minimizes the variance of various $s$-interventional risks: $Var_s(\{\mathcal{R}(h(G), Y|do(S = s))\})$,*

*where the $s$-interventional risk is defined over the $s$-interventional distribution for specific $s \in \mathbb{S}$.*

Guided by the proposed principle, we design the learning strategy of DIR as:

$$\min \mathcal{R}_{\text{DIR}} = \mathbb{E}_s[\mathcal{R}(h(G), Y|do(S = s))] + \lambda \text{Var}_s(\{\mathcal{R}(h(G), Y|do(S = s))\}), \tag{4}$$

where $\mathcal{R}(h(G), Y \mid do(S = s))$ computes the risk under the $s$-interventional distribution, which we will elaborate in Section 2.4. $\text{Var}(\cdot)$ calculates the variance of risks over different $s$-interventional distributions; $\lambda$ is a hyper-parameter to control the strength of invariant learning.

**Justification.** We theoretically justify the DIR principle's ability to discover invariant rationales. Specifically, Theorem 1 shows that the oracle model $f_Y$ respects the DIR principle. Moreover, we suggest that $C$ can be inferred by making the intrinsically interpretable model $h$ conform to the DIR principle under the uniqueness condition (*cf.* Corollary 1). We leave the detailed proofs in Appendix C due to the limited space. By making the distribution-relevant risks indifferent while pursuing low risks, the DIR principle is able to discover the invariant rationales $\tilde{C}$ as the approximation of the oracle rationales $C$, while encouraging $h_{\hat{Y}}$ approaching the oracle model $f_Y$.

## 2.4 DIR-GUIDED IMPLEMENTATION OF INTRINSICALLY-INTERPRETABLE GNNS

With the DIR principle and objective, we present how to implement the intrinsically-interpretable GNNs. We summarize the key notations of this section in Appendix A for clarity. Following Equation 2, a model $h$ with intrinsic interpretability consists of two modules: $h = h_{\hat{Y}} \circ h_{\tilde{C}}$, where $h_{\tilde{C}}$ is to extract a possible rationale, and $h_{\hat{Y}}$ is to make prediction based on the rationale. Moreover, to establish the $s$-interventional distributions, we design an additional module to do the interventions. In a nutshell, our framework consists of four components, as Figure 3 shows.

**Rationale Generator.** It aims to split the input graph instance $g$ into two subgraphs: causal part $\tilde{c}$ and non-causal part $\tilde{s}$. Specifically, given an input graph instance $g = (\mathcal{V}, \mathcal{E})$ with the node set $\mathcal{V}$

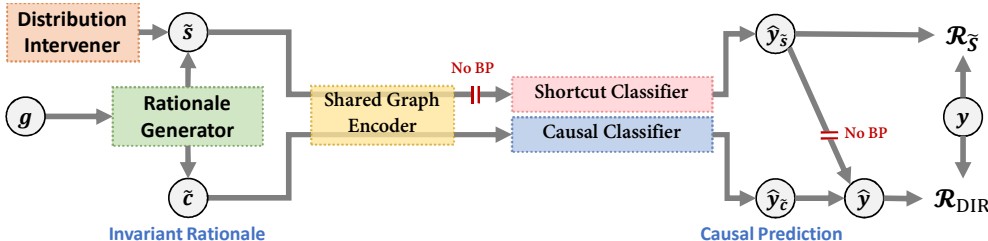

Figure 3: DIR Implementation on GNNs, which includes a rationale generator, a distribution intervener, an encoder and two classifiers. For the inference, we only use $\hat{y}_{\tilde{c}}$ as the prediction.

and the edge set $\mathcal{E}$, its adjacency matrix is $\mathbf{A} \in \{0,1\}^{|\mathcal{V}| \times |\mathcal{V}|}$, where $\mathbf{A}_{ij} = 1$ denotes the edge from node $i$ to node $j$, and $\mathbf{A}_{ij} = 0$ otherwise. The rationale generator first adopts a GNN to generate the mask matrix $\mathbf{M} \in \mathbb{R}^{|\mathcal{V}| \times |\mathcal{V}|}$ on $\mathbf{A}$, where mask $\mathbf{M}_{ij}$ indicates the importance of edge $\mathbf{A}_{ij}$:

$$\mathbf{Z} = \mathrm{GNN}_1(g), \quad \mathbf{M}_{ij} = \sigma(\mathbf{Z}_i^\top \mathbf{Z}_j), \tag{5}$$

where $\sigma(\cdot)$ is the sigmoid function and $\mathbf{Z} \in \mathbb{R}^{|\mathcal{V}| \times d}$ summarizes the $d$-dimensional representations of all nodes. The generator then selects the edges with the highest masks to construct the rationale $\tilde{c}$ and collects $\tilde{c}$'s complement as $\tilde{s}$, as follows:

$$\mathcal{E}_{\tilde{c}} = \mathrm{Top}_r(\mathbf{M} \odot \mathbf{A}), \quad \mathcal{E}_{\tilde{s}} = \mathrm{Top}_{1-r}((1 - \mathbf{M}) \odot \mathbf{A}), \tag{6}$$

where $\mathcal{E}_{\tilde{c}}$ and $\mathcal{E}_{\tilde{s}}$ are the edge sets of $\tilde{c}$ and $\tilde{s}$, respectively; $\mathrm{Top}_r(\cdot)$ selects the top-$K$ edges with $K = r \times |\mathcal{E}|$, and $r$ is the hyper-parameter (*e.g.,* 40%); $\odot$ is the element-wise product. Having obtained the edge sets, we can distill the nodes appearing in the edges to establish $\tilde{c}$ and $\tilde{s}$.

**Distribution Intervener.** It targets at creating interventional distributions. Formally, it first collects the non-causal part of all the instances into a memory bank as $\tilde{\mathbb{S}}$. It next samples a memory $\tilde{s}_i \in \tilde{\mathbb{S}}$ to conduct the intervention $do(S = \tilde{s}_i)$, replacing the complement of the critical subgraph $\tilde{c}_j$ at hand and constructing an intervened pair $(\tilde{c}_j, \tilde{s}_i)$, where $i, j$ are indices.

**Graph Encoder & Classifiers .** Here we represent $h_{\hat{Y}}$ as a combination of a graph encoder and two classifiers. Specifically, it employs another GNN encoder on $\tilde{c}$ to generate node representations $\mathbf{Z}_{\tilde{c}} \in \mathbb{R}^{|\mathcal{V}| \times d}$, and then combines them as graph representation $\mathbf{H}_{\tilde{c}} \in \mathbb{R}^D$ via a global pooling operator, *e.g.,* average pooling. Then it uses a classifier $\Phi_c$ to project the graph representation into a probability distribution over class labels $\hat{y}_{\tilde{c}}$. More formally, the process is as follows:

$$\mathbf{Z}_{\tilde{c}} = \mathrm{GNN}_2(\tilde{c}), \quad \mathbf{H}_{\tilde{c}} = \mathrm{Pooling}(\mathbf{Z}_{\tilde{c}}), \quad \hat{y}_{\tilde{c}} = \Phi_c(\mathbf{H}_{\tilde{c}}). \tag{7}$$

Analogously, we can obtain $\hat{y}_{\tilde{s}}$ for $\tilde{s}$ via the shared encoder and another classifier $\Phi_s$. $\hat{y}_{\tilde{c}}$ is the prediction based merely on the causal part $\tilde{c}$, while $\hat{y}_{\tilde{s}}$ measures the predictive power of the intervened part $\tilde{s}$. Inspired by Cadène et al. (2019), we formulate the joint prediction $\hat{y}$ under the intervention $do(S = \tilde{s})$ as $\hat{y}_{\tilde{c}}$ masked by $\hat{y}_{\tilde{s}}$:

$$\hat{y} = \hat{y}_{\tilde{c}} \odot \sigma(\hat{y}_{\tilde{s}}), \tag{8}$$

where the sigmoid function adjusts the output logits of $\tilde{c}$ to compensate for the spurious biases. In Appendix E, we present examples of how this operation helps discover the causal part.

**Optimization.** Having established the prediction $\hat{y}$ of an instance $g$ under the intervention $do(S = \tilde{s})$, we are capable of getting the $\tilde{s}$-interventional risk similar as Equation 4 as follows:

$$\mathcal{R}(h(G), Y | do(S = \tilde{s})) = \mathbb{E}_{(g,y) \in \mathcal{O}, S = \tilde{s}, C = h_{\tilde{C}}(g)} l(\hat{y}, y), \tag{9}$$

where $(g, y) \in \mathcal{O}$ is a pair of graph instance $g$ and its ground-truth label $y$ from the training set $\mathcal{O}$; $l(\cdot)$ denotes the loss function on a single instance. Moreover, we define the loss for $\Phi_s$ module as:

$$\mathcal{R}_{\tilde{S}} = \mathbb{E}_{(g,y) \in \mathcal{O}, \tilde{s} = g/h_{\tilde{C}}(g)} l(\hat{y}_{\tilde{s}}, y) \tag{10}$$

Specifically, $\mathcal{R}_{\tilde{S}}$ is only backpropagated to the classifier $\Phi_s$ and we set apart the other components from its backpropagation to avoid interference with representation learning. Thus, this loss promotes the $\tilde{S}$-only branch to learn spurious biases given the non-causal features only. Overall, we can jointly optimize these components via the DIR objective and shortcut loss, *i.e.,*

$$\min_{\phi_s} \mathcal{R}_{\tilde{S}} + \min_{\gamma, \theta, \phi_c} \mathcal{R}_{\text{DIR}}. \tag{11}$$

where $\gamma, \theta$ and $(\phi_c, \phi_s)$ are the parameters of the generator, encoder and two classifiers. While in the inference phase, we yield $\tilde{c}$ and $\hat{y}_{\tilde{c}}$ as the causal rationale and the causal prediction of a testing graph $g$, which exclude the influence of the non-causal part $\tilde{s}$.

# 3 EXPERIMENTS

In this section, we conduct extensive experiments to answer the research questions:

- **RQ1:** How effective is DIR in discovering causal features and improving model generalization?
- **RQ2:** What are the learning patterns and insights of DIR training? Especially, how does invariant rationalization help to improve generalization?

## 3.1 SETTINGS

**Datasets.** We use one synthetic dataset and three real datasets of graph classification tasks. Different GNNs are used in different datasets to achieve DIR and early stopping is exploited during training. Here we briefly introduce the datasets, while the details of dataset statistics, deployed GNNs, and training process are summarized in Appendix D.

- **Spurious-Motif** is a synthetic dataset created by following Ying et al. (2019), which involves $18,000$ graphs. Each graph is composed of one base (*Tree*, *Ladder*, *Wheel* denoted by $S = 0, 1, 2$ respectively) and one motif (*Cycle*, *House*, *Crane* denoted by $C = 0, 1, 2$, respectively). The ground-truth label $Y$ is determined by $C$ solely. Moreover, we manually construct false relations of different degrees between $S$ and label $Y$ in the training set. Specifically, in the training set, we sample each motif from a uniform distribution, while the distribution of its base is determined by $P(S) = b \times \mathbb{I}(S = C) + \frac{1-b}{2} \times \mathbb{I}(S \neq C)$. We manipulate $b$ to create Spurious-Motif datasets of distinct biases. In the testing set, the motifs and bases are randomly attached to each other. Besides, we include graphs with large bases to further magnify the distribution gaps.

- **MNIST-75sp** (Knyazev et al., 2019) converts the MNIST images into $70,000$ superpixel graphs with at most 75 nodes each graph. The nodes in the graphs are superpixels, while edges are the spatial distance between the nodes. Every graph is labeled as one of 10 classes. Random noises are added to nodes' features in the testing set.

- **Graph-SST2** (Yuan et al., 2020; Socher et al., 2013) Each graph is labeled by its sentence sentiment and consists of nodes representing tokens and edges indicating node relations. Graphs are split into different sets according to their average node degree to create dataset shifts.

- **Molhiv** (OGBG-Molhiv) (Hu et al., 2020; 2021; Wu et al., 2017) is a molecular property prediction dataset consisting of molecule graphs, where nodes are atoms, and edges are chemical bonds. Each graph is labeled according to whether a molecule inhibits HIV replication or not.

**Baselines.** We thoroughly compare DIR with Empirical Risk Minimization (ERM) and two classes of baselines:

- **Interpretable Baselines:** Graph Attention (Veličković et al., 2018) and graph pooling operations including ASAP (Ranjan et al., 2020), Top-$k$ Pool (Gao & Ji, 2019) and SAG Pool (Lee et al., 2019). We use their generated masks on graph structures as rationales. We also include GSN (Bouritsas et al., 2020), a topologically-aware message passing scheme which enriches GNNs with interpretable structural features.

- **Robust/Invariant Learning Baselines:** Group DRO (Sagawa et al., 2019), IRM (Arjovsky et al., 2019), V-REx (Krueger et al., 2021). This class of algorithms improves the robustness and generalization for GNNs, which helps the models better generalize in unseen groups or out-of-distribution datasets. We use random groups or partitions during the model training.

We also include an ablation model of DIR, DIR-Var, which sets $\lambda = 0$, *i.e.,* discards the variance term in $\mathcal{R}_{\text{DIR}}$, to show the effectiveness of the variance regularization in the DIR objective.

**Metrics.** We use ROC-AUC for Molhiv and ACC for the other three datasets. Moreover, for Spurious-Motif dataset, we use the precision metric to evaluate the coincidence between model rationales and the ground-truth rationales, and validate the interpretability ability quantitatively.

Table 1: Performance on the Synthetic Dataset and Real Datasets. In Spurious-Motif dataset, we color brown for the results lower than ERM, where $b$ is the indicator of the confounding effect.

| | Spurious-Motif | | | | MNIST-75sp | Graph-SST2 | Molhiv |
| | Balance | $b = 0.5$ | $b = 0.7$ | $b = 0.9$ | | | |
|---|---|---|---|---|---|---|---|
| ERM | $42.99_{\pm1.93}$ | $39.69_{\pm1.73}$ | $38.93_{\pm1.74}$ | $33.61_{\pm1.02}$ | $12.71_{\pm1.43}$ | $81.44_{\pm0.59}$ | $76.20_{\pm1.14}$ |
| Attention | $43.07_{\pm2.55}$ | $39.42_{\pm1.50}$ | $37.41_{\pm0.86}$ | $33.46_{\pm0.43}$ | $15.19_{\pm2.62}$ | $81.57_{\pm0.71}$ | $75.84_{\pm1.33}$ |
| ASAP | $44.44_{\pm8.19}$ | $44.25_{\pm6.87}$ | $39.19_{\pm4.39}$ | $31.76_{\pm2.89}$ | $15.54_{\pm1.87}$ | $81.57_{\pm0.84}$ | $73.81_{\pm1.17}$ |
| Top-$k$ Pool | $43.43_{\pm8.79}$ | $41.21_{\pm7.05}$ | $40.27_{\pm7.12}$ | $33.60_{\pm0.91}$ | $14.91_{\pm3.25}$ | $79.78_{\pm1.35}$ | $73.01_{\pm1.65}$ |
| SAG Pool | $45.23_{\pm6.76}$ | $43.82_{\pm6.32}$ | $40.45_{\pm7.50}$ | $33.60_{\pm1.18}$ | $14.31_{\pm2.44}$ | $80.24_{\pm1.72}$ | $73.26_{\pm0.84}$ |
| GSN | $43.18_{\pm5.65}$ | $34.67_{\pm1.21}$ | $34.03_{\pm1.69}$ | $32.60_{\pm1.75}$ | $19.03_{\pm2.39}$ | $82.54_{\pm1.16}$ | $74.53_{\pm1.90}$ |
| Group DRO | $41.51_{\pm1.11}$ | $39.38_{\pm0.93}$ | $39.32_{\pm2.23}$ | $33.90_{\pm0.52}$ | $15.13_{\pm2.83}$ | $81.29_{\pm1.44}$ | $75.44_{\pm2.70}$ |
| V-REx | $42.83_{\pm1.59}$ | $39.43_{\pm2.69}$ | $39.08_{\pm1.56}$ | $34.81_{\pm2.04}$ | $18.92_{\pm1.41}$ | $81.76_{\pm0.08}$ | $75.62_{\pm0.79}$ |
| IRM | $42.26_{\pm2.69}$ | $41.30_{\pm1.28}$ | $40.16_{\pm1.74}$ | $35.12_{\pm2.71}$ | $18.62_{\pm1.22}$ | $81.01_{\pm1.13}$ | $74.46_{\pm2.74}$ |
| DIR-Var | $45.87_{\pm2.61}$ | $43.81_{\pm1.93}$ | $42.69_{\pm1.77}$ | $37.12_{\pm1.56}$ | $17.74_{\pm4.17}$ | $81.74_{\pm0.89}$ | $76.05_{\pm0.86}$ |
| **DIR** | $\mathbf{47.03_{\pm2.46}}$ | $\mathbf{45.50_{\pm2.15}}$ | $\mathbf{43.36_{\pm1.64}}$ | $\mathbf{39.87_{\pm0.56}}$ | $\mathbf{20.36_{\pm1.78}}$ | $\mathbf{83.29_{\pm0.53}}$ | $\mathbf{77.05_{\pm0.57}}$ |

Table 2: Precision@5 on Spurious-Motif.

| Model | Balance | $b = 0.5$ | $b = 0.7$ | $b = 0.9$ |
|---|---|---|---|---|
| Attention | $0.183_{\pm0.018}$ | $0.183_{\pm0.130}$ | $0.182_{\pm0.014}$ | $0.134_{\pm0.013}$ |
| ASAP | $0.187_{\pm0.030}$ | $0.188_{\pm0.023}$ | $0.186_{\pm0.027}$ | $0.121_{\pm0.021}$ |
| Top$k$ Pool | $0.215_{\pm0.061}$ | $0.207_{\pm0.057}$ | $0.212_{\pm0.056}$ | $0.148_{\pm0.018}$ |
| SAG Pool | $0.212_{\pm0.033}$ | $0.198_{\pm0.062}$ | $0.201_{\pm0.064}$ | $0.136_{\pm0.014}$ |
| **DIR** | $\mathbf{0.257_{\pm0.014}}$ | $\mathbf{0.255_{\pm0.016}}$ | $\mathbf{0.247_{\pm0.012}}$ | $\mathbf{0.192_{\pm0.044}}$ |

## 3.2 Main Results (RQ1)

To fairly compare the methods, we train each model under the same training settings as described in Appendix D. The overall results are summarized Table 1, and we have the following observations:

1. **DIR has better generalization ability than the baselines.** DIR outperforms the baselines consistently by a large margin. Specifically, for MNIST-75sp dataset, DIR surpasses ERM by 7.65% and ASAP by 4.82%. Although structure features are shown to be helpful in mitigating feature distribution shift, DIR still performs better than GSN. For Graph-SST2 and Molhiv, DIR achieves the highest performance with low variance. For Spurious-Motif, DIR outstrips IRM averagely by 4.23% and SAG by 3.16% across different degrees of spurious bias. Such improvements strongly validate that DIR can generalize better in various environments.

2. **DIR is consistently effective under different bias degrees, while the baselines easily fail.** For interpretable baselines, Attention fails to make salient improvements when bias exists, and pooling methods also fall through under severe bias. This is empirically in line with our presumption that GNNs are easily biased to latch on spurious relations or non-causal features and thus generalize poorly in OOD data. For robust/invariant learning baselines, IRM underperforms ERM when $b$ is small. This evidence is accordant with the conclusion in Ahuja et al. (2021) that IRM is guaranteed to be close to the desired OOD solutions when confounders exist, while it has no obvious advantage to ERM under covariate shift. Moreover, Group DRO and V-REx follow a similar pattern. In contrast, DIR works well in various scenarios. We credit such reliability to the rationales discovery from which the causal features $C$ are potentially extracted, and the relation $C \rightarrow Y$ learned by the GNNs is invariant across the distribution changes in the testing set.

3. **Data augmentation by intervention is beneficial while the variance regularization further boosts model performance.** Interestingly, the ablation model DIR-Var has already exceeded some of the baselines. We attribute such improvement to data augmentation via interventional distributions. On top of DIR-Var, DIR improves the model performance by averagely 1.57% in Spurious-Motif and 2.62% in MNIST-75sp. This suggests that the variance regularization demands a stronger invariance condition and is instructive for searching causal features.

4. **DIR has better intrinsic interpretability than the baselines.** In Table 2, we report intrinsic interpretable models' performance *w.r.t.* Precision@5. From the consistent improvements over the baselines, we find DIR has an advantage in discovering causal features. And the performance gap between DIR and the baselines becomes more significant when the bias increases.

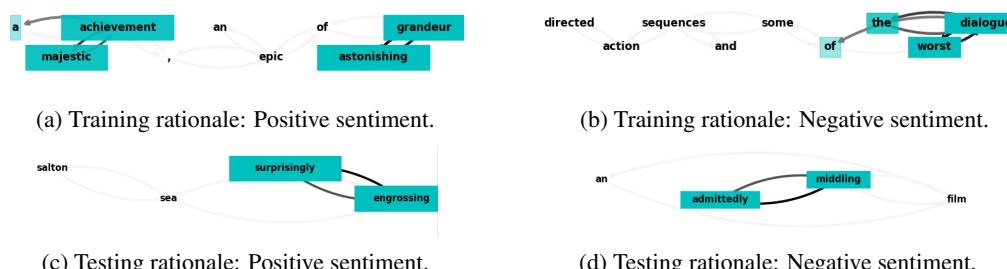

(a) Training rationale: Positive sentiment.

(b) Training rationale: Negative sentiment.

(c) Testing rationale: Positive sentiment.

(d) Testing rationale: Negative sentiment.

Figure 4: Visualization of DIR Rationales. Each graph shows a comment, *e.g.,* "*a majestic achievement, an epic of astonishing grandeur*" in (a), where rationales are highlighted by deep colors.

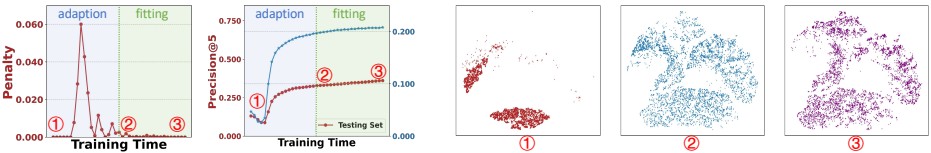

(a) The first two subfigures show the training curves *w.r.t.* variance penalty and precision, on Spurious-Motif. The last three subfigures present the rationale distributions of the inspection points, which are visualized by t-SNE (van der Maaten, 2008).

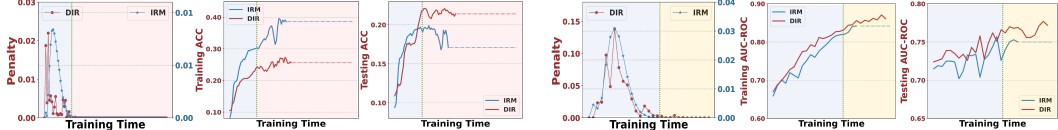

(b) The first three subfigures present the training curves *w.r.t.* variance penalty and ACC on MNIST-75sp, while the last three illustrate the curves *w.r.t.* variance penalty and AUC-ROC on Molhiv.

Figure 5: Two-stage Training Dynamics of DIR.

## 3.3 IN-DEPTH STUDY (RQ2)

We empirically analyze the DIR's properties which hopefully give insights into its mechanisms and can be instructive for the existing training paradigms of deep models.

**Rationale Visualization.** Towards an intuitive understanding of DIR, we first present some cases of the discovered rationale for Graph-SST2 in Figure 4. DIR is able to emphasize the tokens that directly result in the sentences' positive or negative sentiments, which are reliable and faithful rationales. Specifically, DIR highlights the positive words *"majestic achievement"* and *"astonishing grandeur"* in Figure 4a and underscores the negative words *"worst dialogue"* in Figure 4b as the rationales, which are clearly salient for the positive and negative sentiments, respectively. Furthermore, DIR can focus persistently on the causal features for OOD testing data. For example, it selects *surprisingly engrossing* and *"admittedly middling"* in Figures 4c and 4d, respectively. This again validates the effectiveness of DIR: (1) $h_{\tilde{C}}$ is well-learned to distinguish causal and non-causal features under various interventional distributions; and (2) $h_{\tilde{Y}}$ conducts message-passing on the highlighted rationales, extracts the graph representations, and finally outputs the predictions with high accuracy. See Appendix F.1 for more examples in Graph-SST2 and Spurious-Motif datasets.

**Two-stage Training Dynamics.** As Figure 5a displays, we find a pattern from the Var-Time curve — during training DIR, the variance penalty (*i.e.,* Var$_s$ in Equation 4) first increases and then decreases to almost zero. Moreover, there exists an interesting correlation between the variance penalty and the precision metrics — that is, the precision rises dramatically as the penalty increases while growing slowly as the penalty decreases. To probe this learning pattern, we further visualize the rationale distribution in three turning points: (1) the start, (2) the middle, and (3) the end of training. Interestingly, the rationale distribution at the middle point is highly similar to that at the ending point. This illustrates two stages, adaption and fitting, in the patterns. By "adaption", we mean that the exhibition of $h_{\tilde{C}}$, *i.e.,* learning to select salient feature $\tilde{C}$, is mainly conducted during the initial training stage. Since the penalty value can be seen as the magnitude to violate the invariance condition, this stage explores the rationales that satisfy the DIR principle. Correspondingly, $h_{\tilde{Y}}$ adapts

quickly with the input of varying rationales generated by $h_{\tilde{C}}$. By "fitting", we mean that, in the later training process, $h_{\tilde{C}}$ only makes small changes, resulting in the substantially unchanged rationales compared to the initial training process, which is learned from the rationale generator to conform to the DIR principle. This could also imply that based on the well-learned rationales, DIR mainly optimizes $h_{\tilde{Y}}$ to consolidate the functional relation $\tilde{C} \to Y$ until model convergence.

Moreover, we compare the learning patterns of IRM and DIR in Figure 5b, where the penalty term of IRM (the gradient norm penalty in IRMv1 (Arjovsky et al., 2019)) follows a similar pattern to the DIR penalty. Notably, in MNIST-75sp, while IRM consistently outperforms DIR *w.r.t.* Training ACC, it does not improve and even degrades the performance in the testing dataset due to over-fitting. However, DIR shows the solid resistance for over-fitting, partly thanks to the valid rationales exhibited in the adaption stage. For Molhiv, DIR outperforms IRM as the rationales filter out irrelevant or spurious structures bootless for classification tasks and are beneficial for generalization.

**Sensitivity Analysis.** We conduct a sensitivity analysis of model performance *w.r.t.* $\lambda$ in Appendix F.2, which shows that DIR surpasses the best baselines under a relatively large range of $\lambda$.

## 4 RELATED WORKS

**Inherent Interpretability of GNNs.** We summarize two classes of the existing methods to build deep interpretable GNNs, (i) Attention (Vaswani et al., 2017; Veličković et al., 2018), which can be broadly interpreted as importance weights on representations.(ii) Pooling (Lee et al., 2019; Knyazev et al., 2019; Gao & Ji, 2019), which selectively performs down-sampling on representations. We include it in this category when it involves selection importance. However, the mechanisms to generate the rationales could be epistemic, as they only reflect the probabilistic relations between data and predicted labels (Pearl, 2000), which may not hold true in all data distributions. Thus, the rationales could fail to align with causal features and even degrade model performance due to being "fooled" by spurious features (Chang et al., 2020).

**Invariant Learning.** Backed by causal theory, invariant learning assumes the causal relation from the causal factors $C$ to the response variable $Y$ remains invariant unless we intervene on $Y$. As the most prevailing formulation, IRM (Arjovsky et al., 2019) extends the invariance assumption from feature level to representation level and finds a data representation $\Phi$ such that $\Omega \circ \Phi$ matches for all environments, where $\Omega$ is the classifier. However, concerns about its feasibility (Rosenfeld et al., 2021; Ahuja et al., 2021) and optimality (Kamath et al., 2021) have been discussed recently. Besides IRM, variance penalization across environments is shown to be effective for recovering invariance (Krueger et al., 2021; Xie et al., 2020; Teney et al., 2020). Notably, the existing methods generally require accessing different environments, thus additionally involving environment inference (Creager et al., 2021; Wang et al., 2021b). Similarly motivated as ours, Chang et al. (2020) discover rationales $Z$ by minimizing the performance gap between environment-agnostic predictor $f(Z)$ and environment-aware predictor $f(Z, E)$. In graph domain, Bevilacqua et al. (2021) construct graph representations from subgraph densities and use attribute symmetry regularization to mitigate the shift of graph size and vertex attribute distributions.

## 5 CONCLUSION & FUTURE WORK

In this work, we rigorously study the intrinsic interpretability of Graph Neural Networks from a causal perspective. Our concerns are towards the exhibition of shortcut features when generating the rationales. And we proposed an invariant learning algorithm, DIR, to discover the causal features for rationalization. The core of DIR lies in the construction of environments (*i.e.,* interventional distributions) and thus distilling the salient features as rationales that are consistently informative and uniform across these environments. Such rationales serve as the probing towards model mechanisms and are demonstrated to be effective in generalization. In the experiments, we highlight an adaption-fitting training dynamics for DIR to reveal its learning pattern. In the future, we will build more reliable and expressive interpretable models that are feasible under various assumptions, which potentially calls for high-level interpretability. We recommend interested readers go to the open discussion in Appendix G for the detailed description.

## ACKNOWLEDGMENT

This work was supported by the National Key Research and Development Program of China (2020AAA0106000), the National Natural Science Foundation of China (U19A2079), the Sea-NExT Joint Lab, and Singapore MOE AcRF T2.

## ETHICS STATEMENT

In this work, we propose a novel algorithm for intrinsic interpretable models, where no human subject is related. This synthetic dataset is made available in the anonymous link (*cf.* Section 3.1). We believe the exhibition of rationales is beneficial for inspecting and eliminating potential discrimination and fairness issues in deep models for real applications.

## REPRODUCIBILITY STATEMENT

We summarize the efforts made to ensure reproducibility in this work. (1) Datasets: We use one synthetic dataset which is made available (*cf.* the anonymous link in Section 3.1), and three public datasets where the processing details are included in Appendix D. (2) Model Training: We provide the procedure of training in Algorithm A and the training details (including hyper-parameter settings) in Appendix D which are consistent with our implementation in the code (*cf.* the anonymous link in Section 3.1). (3) Theoretical Results: All assumptions and proofs can be referred to Appendix C.

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

# A    NOTATIONS & ALGORITHM

Key Notations in the Main Paper.

| Symbol | Definition |
|--------|-----------|
| $g$ | graph instance |
| $c$ / $s$ | ground truth causal or confounding subgraph |
| $\tilde{c}$ / $\tilde{s}$ | generated rationale or complement of rationale instance |
| $C$ / $S$ | variables in the causal graph |
| $\mathbb{S}$ / $\tilde{\mathbb{S}}$ | space of the ground truth or identified spurious features |
| $\hat{y}_{\tilde{c}}$ / $\hat{y}_{\tilde{s}}$ | causal or spurious prediction |
| $\hat{y}$ | joint prediction |
| $h_{\tilde{C}}$ | rationale generator |
| $\Phi_1$ / $\Phi_2$ | causal or spurious classifier |

---

**Algorithm 1** Pseudocode for DIR in training interpretable Graph Neural Networks (Batch Version)

---

**Require:** Training data distribution $\mathcal{P}_{tr}(G)$; number of classes $Q$; Stepsize $\alpha$; hyper-parameter $\lambda$

1: Randomly initialize the parameters of generator $h_{\tilde{C}}$, encoder $h_\theta$ (includes GNN$_2$ and Pooling layer), two classifiers $\Phi_1$ and $\Phi_2$, which are denoted as $\gamma, \theta, \phi_1, \phi_2$, respectively.

2: **while** not converge **do**

3:     Sample graphs $\{(g^i, y^i)\}_{i=1}^B$ from $\mathcal{P}_{tr}(G)$

4:     Generate each rationale and its complement: $(\tilde{c}^i, \tilde{s}^i) \leftarrow h_{\tilde{C}}(g^i)$, for $i = 1, \dots, B$.

5:     **for** each $\tilde{s}^i$ **do**

6:         Intervener $h_I$ operates $do(S = \tilde{s}^i)$

7:         Model forward: $\hat{y}_{\tilde{s}} = \Phi_2(h_\theta(\tilde{s}^i)) \in \mathbb{R}^{1 \times Q}$, $\{\hat{y}_{\tilde{c}}\}_{i=1}^B = \Phi_1(h_\theta(\{\tilde{c}^i\}_{i=1}^B)) \in \mathbb{R}^{B \times Q}$

8:         `# block BP of DIR risk to shortcut branch`
           Obtain joint prediction $\hat{y} = [\hat{y}^1, \dots, \hat{y}^B]$, where $\hat{y}^j = \hat{y}_{\tilde{c}} \odot \sigma(\hat{y}_{\tilde{s}}^j).\text{detach}()$

9:         Compute and record risk $\mathcal{R}(\hat{y}_{\tilde{s}}, y^i)$

10:        Compute and record $\tilde{s}^i$-interventional risk.

11:    **end for**

12:    Compute $\mathcal{R}_{\text{DIR}}$ via Eq. 4 and $\mathcal{R}_{\tilde{S}}$ via Eq. 11

13:    Update parameters: $\phi_2 = \phi_2 - \alpha \nabla_{\phi_2} \mathcal{R}_{\tilde{S}}$; $\phi_1 = \phi_1 - \alpha \nabla_{\phi_1} \mathcal{R}_{\text{DIR}}$;
           $\gamma = \gamma - \alpha \nabla_\gamma \mathcal{R}_{\text{DIR}}$; $\theta = \theta - \alpha \nabla_\theta \mathcal{R}_{\text{DIR}}$

14: **end while**

---

# B    INSTANTIATED CAUSAL GRAPHS

We instantiate possible causal graphs in Figure 2a. Specifically, we use the example of Base-Motif graphs, whose labels are determined by the motif types. We use $C = 0, 1, 2$ to denote cycle, house, crane, respectively; And use $S = 0, 1, 2$ to denote ladder, tree, wheels, respectively.

- $C \perp\!\!\!\perp S$: Base graphs and motif graphs are independently sampled and attached to each other.
- $C \rightarrow S$: Type of each motif respects to a given (static) probability distribution. According to the value of $C$, the probability distribution of its base graph is given by

$$P(S) = \begin{cases} 0.6 & \text{if } S = C \\ 0.2 & \text{otherwise} \end{cases} \tag{12}$$

- $S \rightarrow C$: Similar to the example for $C \rightarrow S$.
- $S \leftarrow E \rightarrow C$: Suppose there is a latent variable $E$ takes continuous value from 0 to 1. Then the probability distribution of $S$ and $C$ s.t.

$$S \sim \mathcal{B}(3, E) \quad C \sim \mathcal{B}(3, 1 - E) \tag{13}$$

where $\mathcal{B}$ stands for binomial distribution, *i.e.*, for variable $X$, if $X \sim \mathcal{B}(n, p)$, then we have

$$P(X = k \mid p, n) = \begin{pmatrix} n \\ k \end{pmatrix} p^k (1 - p)^{n-k}$$

## C  THEORY

### C.1  ASSUMPTION

We phrase the SCM in Figure 2a as the following assumption:

**Assumption 1 (Invariant Rationalization (IR))** *There exists a rationale $C \subseteq G$, such that the structural equation model*

$$Y \leftarrow f_Y(C, \epsilon_Y), \epsilon_Y \perp\!\!\!\perp C$$

*and the probability relation*

$$S \perp\!\!\!\perp Y \mid C$$

*hold for every distribution $\tilde{\mathcal{P}}$ over $\mathcal{P}(G, Y)$, where $S$ denotes the complement of $C$. Also, we denote $f_Y$ as the oracle structural equation model.*

By "oracle", we mean that $f_Y$ is the perfect structure equation model, which, when $C$ is available, predicts the response variable with the minimum expected loss over any distribution $\tilde{\mathcal{P}}$. Or formally,

$$f_Y := \arg\min_f \mathcal{R}(f) = \arg\min_f \mathbb{E}_{(G,Y) \sim \tilde{\mathcal{P}}, \epsilon_Y}[l(f(C, Y), \epsilon_Y), Y)]. \tag{14}$$

where $l$ is the task-specific loss function and we ignore the exogenous noise $\epsilon_Y$ in $f_Y$'s input except as otherwise noted.

Next, we argue that the assumption is commonly satisfied. For example, for sentences labeled by sentiment, $C$ can represent the positive/negative words that cause the sentiment, while $S$ includes the prepositions and linking words. For molecule graphs labeled by specific properties, $C$ and $S$ can represent the functional groups and carbon structures, respectively. Note that IR assumption enables and calls the introduction of interpretability, highlighting salient features and exhibiting human accessible checks. More importantly, it guarantees the model performance under possible feature reduction, *i.e.,* $C \subset G$.

We also see cases going beyond the IR Assumption. For example, $G$ could be a generic function of $S$ and $C$, instead of a simple joint. We use a toy example to elaborate this point. Following the Spurious-Motif dataset, we assume each graph has multiple motifs (house, cycle, crane) with only one type and is labeled by the motif type. Thus, the causal feature $C$ will be the motifs. Let the spurious feature $S$ be "the way we connect the motifs". For example, we can place the house motifs in a queue sequence and connect the adjacent motifs, thus forming the graph in a "line" shape. Or we can place the houses in a cycle order and connect them into a ring. We further make such graph structures strongly correlated with the motif types. Thus, individual $S$ and $C$ may be intractable individually in the feature level. For example, if we separate the cycle-shaped houses into two lines, the spurious pattern could be broken while the part of the causal feature would be lost. In other words, $S$ and $C$ are dependent variables. Thus, they can't be extracted and modeled separately, which goes out of the scope of our work.

Given that $S$ and $C$ are separable, we further make the following assumption to avoid the confusion of $S$ and $C$:

**Assumption 2 (Feature Induction)** *Define power set operation as $\mathcal{P}^*(\cdot)$. For data $G = S \cup C$ and label $Y$, if $S \perp\!\!\!\perp Y \mid C$ holds for any distribution $\tilde{\mathcal{P}}$ over $\mathcal{P}(G, Y)$, then it implies that for any induced feature $S' \in \mathcal{P}^*(S)$, we have $S' \perp\!\!\!\perp Y \mid C$ holds for the distribution $\tilde{\mathcal{P}}$.*

This assumption also implies that $C$ could not be induced by $S$ when $|C| \leq |S|$. Thus, any feature subset $C'$ except for $C$ would violate the conditional independence condition. For images, this assumption is natural for the splicing of $S$ doesn't typically change its semantics. For example, the splicing of land background would still be divided land. While for graphs, here we assume the causal subgraph's uniqueness among the induced complement graphs.

### C.2  PROOFS

**Theorem 1 (Necessity)** *Suppose $S \rightarrow C$ does not exist, then the oracle function $f_Y$ satisfies the DIR Principle (where $C$ is given) over every distribution $\tilde{\mathcal{P}} \in \mathcal{P}(G, Y)$.*

**Proof:** We first prove the fact that $P(Y = y \mid do(S = s)) = P(Y = y)$ for distribution $\tilde{\mathcal{P}}$. Specifically, we use $P_I^{(s)}$ to denote the s-interventional distribution.

- If $C \to S$,

$$P(Y = y \mid do(S = s)) \xlongequal{\text{by definition}} P_I^{(s)}(Y = y \mid S = s)$$

$$= \sum_c P_I^{(s)}(Y = y \mid S = s, C = c)P_I^{(s)}(C = c \mid S = s)$$

$$\xlongequal{\text{given } C \to S} P_I^{(s)}(Y = y \mid S = s, C = c)P_I^{(s)}(C = c)$$

$$\xlongequal{\text{given } (Y \perp\!\!\!\perp S|C)_{\tilde{\mathcal{P}}}} \sum_c P_I^{(s)}(Y = y \mid C = c)P_I^{(s)}(C = c)$$

$$\xlongequal{\text{given invariance condition}} \sum_c P(Y = y \mid C = c)P(C = c)$$

$$= P(Y = y)$$

- If $C \perp\!\!\!\perp S$,

$$P(Y = y \mid do(S = s)) \xlongequal{\text{by definition}} P_I^{(s)}(Y = y \mid S = s)$$

$$\xlongequal{\text{given } S \text{ has no endogenous parent}} P(Y = y \mid S = s)$$

$$\xlongequal{\text{given} C \perp\!\!\!\perp S} \sum_c P(Y = y \mid C = c, S = s)P(C = c)$$

$$= \sum_c P(Y = y \mid C = c, S = s)P(C = c \mid S = s)$$

$$\xlongequal{\text{given } (Y \perp\!\!\!\perp S|C)_{\tilde{\mathcal{P}}}} \sum_c P(Y = y \mid C = c)P(C = c)$$

$$= P(Y = y)$$

- If $C \leftarrow E \to S$,

$P(Y = y \mid do(S = s))$

$\xlongequal{\text{by definition}} P_I^{(s)}(Y = y \mid S = s)$

$\xlongequal{\text{given } E \to S} \sum_e P_I^{(s)}(Y = y \mid S = s, E = e)P_I^{(s)}(E = e)$

$= \sum_e \sum_c P_I^{(s)}(Y = y \mid S = s, E = e, C = c)P_I^{(s)}(C = c|S = s, E = e)P_I^{(s)}(E = e)$

$\xlongequal{\text{given } (Y \perp\!\!\!\perp \{S,E\}|C) \text{ and } (C \perp\!\!\!\perp S|E)} \sum_e \sum_c P_I^{(s)}(Y = y \mid C = c)P_I^{(s)}(C = c|E = e)P_I^{(s)}(E = e)$

$= \sum_e \sum_c P(Y = y \mid C = c)P(C = c|E = e)P(E = e)$

$= \sum_e \sum_c P(Y = y \mid C = c, E = e)P(C = c|E = e)P(E = e)$

$= P(Y = y)$

As $P(Y = y \mid do(S = s)) = P(Y = y)$ holds true for every distribution $\tilde{\mathcal{P}}$, which is invariant *w.r.t.* iterative variable $S$. Moreover, we have $P(C = c \mid do(S = s)) = P_I^{(s)}(C = c) = P(C = c)$. This

indicates that the intervention on $S$ leave the causal structure $C \rightarrow Y$ untouched. Thus, we have

$$
\begin{aligned}
\text{Var}\left(\{\mathcal{R}(f_Y \mid do(s)) \mid s \in \mathbb{S}\}\right) &= \text{Var}\left(\left\{\mathbb{E}_{(G,Y) \sim P_I^{(s)}(G,Y), C \subset G}[l(f_Y(C), Y)] \mid s \in \mathbb{S}\right\}\right) \\
&= \text{Var}\left(\left\{\mathbb{E}_{(C,Y)}[l(f_Y(C), Y)] \mid s \in \mathbb{S}\right\}\right) \\
&= 0
\end{aligned}
$$

Finally, taking the definition of $f$, we have

$$
\begin{aligned}
f_Y &= \arg\min_f \mathbb{E}_{s \in \mathbb{S}}\left[\mathbb{E}_{(G,Y) \sim P_I^{(s)}(G,Y), C \subset G}[l(f(C), Y)]\right] \\
&= \arg\min_f \mathbb{E}_{s \in \mathbb{S}}\left[\mathbb{E}_{(C,Y)}[l(f(C), Y)]\right] \\
&= \arg\min_f \mathbb{E}_{s \in \mathbb{S}}[\mathcal{R}(f \mid do(s))]
\end{aligned}
$$

Hence, $f_Y$ takes the minimum penalty and satisfies the DIR Principle. $\qquad\square$

Notably, if $S \rightarrow C$, then $\text{Var}\left(\{\mathcal{R}(f_Y \mid do(s)) \mid s \in \mathbb{S}\}\right)$ may not equal to zero since $c \sim P_I^{(s)}(C \mid S = s)$. In such case, $f_Y$ is not necessarily satisfied to DIR Principle. That is, although $f_Y$ still minimizes $\mathcal{R}(f \mid do(S))$, we can't be sure whether it reaches the lower bound of $\text{Var}\left(\{\mathcal{R}(f_Y \mid do(s)) \mid s \in \mathbb{S}\}\right)$ without knowledge about the specific data distribution. Thus, we only consider the cases of $C \rightarrow S$, $C \perp\!\!\!\perp S$ and $C \leftarrow E \rightarrow S$ in the following discussion.

**Theorem 2 (Uniqueness)** *Suppose $l$ is a strict loss function and there exists one and only one non-trivial subset $C$, then there exists a unique structure equation model $f_Y$ s.t. it satisfies the DIR Principle.*

**Proof:** Since $f_Y$ exists and satisfies the DIR Principle, we only need to prove its uniqueness under the given conditions. Otherwise, suppose we have another structure equation $f_Y' \neq f_Y$ satifies the DIR Principle. Specifically, there exists a datum $(g, y)$ s.t. $f_Y'(c) \neq f_Y(c)$. Thus, we have $l(f_Y'(c), y) > l(f_Y(c), y)$. Given that $\text{Var}\left(\{\mathcal{R}(f_Y' \mid do(s)) \mid s \in \mathbb{S}\}\right) \geq 0 = \text{Var}\left(\{\mathcal{R}(f_Y \mid do(s)) \mid s \in \mathbb{S}\}\right)$, we have $\mathcal{R}_{\text{DIR}}(f_Y') > \mathcal{R}_{\text{DIR}}(f_Y)$. $\qquad\square$

In reality, there could be multiple candidates of $C$, *e.g.*, $C_i, C_j$ s.t. $\mathcal{R}_{\text{DIR}}(f_Y^{(C_i)}) = \mathcal{R}_{\text{DIR}}(f_Y^{(C_j)})$, where $f_Y^{(C_i)}$ is the structure equation corresponds to $C_i$. Thus, it calls for the selection of $C$ to avoid the learning of suboptimal $f_Y$. Inspired by Occam's Razor, we define

$$
C^* = \arg\min |C| \tag{15}
$$

as the preferred rationale, or rationale of parsimony. We argue that rationales are not to be extended beyond necessity, which poses simpler hypotheses about causality. As the search of $C^*$ is NP-hard (the worst time complexity is exponential), we use fixed size for the learned rationales in our experiments and leave a better optimization to future work.

**Corollary 1 (Necessity and Sufficiency)** *Suppose $l$ is a strict loss function and there exists one and only one non-trival subset $C$, then any structure causal model $f_Y'$ s.t. it satisfies the DIR Principle iff. $f_Y' = f_Y$.*

This is directly obtained from Theorem 2. Thus, under the unique constraint of $C$, we can approach the oracle $f_Y$ by optimizing the DIR objective, which maintains the invariant causal relation between the causal feature and the response variable $Y$. In another way, based on the uniqueness of the feasible rationale, the optimization of the DIR Principle on the intrinsic interpretable model $h$ (where $C$ is exhibited inside of $h$) pushes the approach to $C$ with rationales $\tilde{C}$. Then, $f_Y$ can also be approached as an invariant predictor based on the learning from $\tilde{C}$.

# D  SETTING DETAILS

Table 3: **Statistics of Graph Classification Datasets.**

|  | Spurious-Motif | | | MNIST-75sp (reduced) | | | Graph-SST2 | | | OGBG-Molhiv | | |
|---|---|---|---|---|---|---|---|---|---|---|---|---|
|  | Train | Val | Test | Train | Val | Test | Train | Val | Test | Train | Val | Test |
| Classes# | | 3 | | | 10 | | | 2 | | | 2 | |
| Graphs# | 9,000 | 3,000 | 6,000 | 20,000 | 5,000 | 10,000 | 28,327 | 3,147 | 12,305 | 32,901 | 4,113 | 4,113 |
| Avg. N# | 25.4 | 26.1 | 88.7 | 66.8 | 67.3 | 67.0 | 17.7 | 17.3 | 3.45 | 25.3 | 27.79 | 25.3 |
| Avg. E# | 35.4 | 36.2 | 131.1 | 539.3 | 545.9 | 540.4 | 33.3 | 33.5 | 4.89 | 54.1 | 61.1 | 55.6 |
| Backbone | Local Extremum GNN (Ranjan et al., 2020) | | | $k$-GNNs (Morris et al., 2019) | | | ARMA (Bianchi et al., 2019) | | | GIN + Virtual nodes (Xu et al., 2019; Hu et al., 2021) | | |
| Neuron# | [4,32,32,32] | | | [5,32,32,32] | | | [768,128,128,2] | | | [9,300,300,300,1] | | |
| Global Pool | global mean pool | | | global max pool | | | global mean pool | | | global add pool | | |
| Gen. Type | Scale & Correlation Shift | | | Noise | | | Degree & Scale Shift | | | / | | |

**Datasets**  We summarize dataset statistics in Table 3, and introduce the node/edge features and the preprocessing in each datasets:

- **Spurious-Motif.** We use random node features and constant edge weights in this dataset.

- **MNIST-75sp.** The nodes in the graphs are superpixels, and node features are the concatenation of pixel intensities (RGB channels) and coordinates of their mass centers. Edges are the spatial distance between the superpixel centers, while we filter the edges with a distance less than 0.1 to make the graphs sparser.

- **Graph-SST2.** We use constant edge weight and filter the graphs with edges less than three. We initialize the node features by the pre-trained BERT (Devlin et al., 2018) word embedding.

- **OGBG-Molhiv.** We use the official released dataset in our experiment.

**GNNs.**  We summarize the backbone GNNs for each dataset in Table 3. The number of neurons in the sequent layers (in forwarding order) is reported. We use ReLU as activation layers and different global pooling layers. In OGBG-Molhiv, we adopt one fully connected layer for the prediction layers while using two fully connected layers for the models in other datasets. For baselines with node pooling/node attention, we add one node pooling/attention layer in the second convolution layer.

**Training Optimization & Early Stopping.**  All experiments are done on a single Tesla V100 SXM2 GPU (32 GB). During training, we use Adam (Kingma & Ba, 2015) optimizer. The maximum number of epochs is 400 for all datasets. We use Stochastic Gradient Descent (SGD) for the optimization on Graph-SST2 and OGBG-Molhiv and Gradient Descent (GD) for the other two datasets. Also, we exhibit early stopping to avoid overfitting of the training dataset. Specifically, in MNIST-75sp, Graph-SST2 and OGBG-Molhiv, each model is evaluated on a holdout in-distribution validation dataset after each epoch. While for Spurious-Motif, we use an unbiased validation dataset (*i.e.,* without spurious relations compared to the training dataset). If the model's performance on the validation dataset is without improvement (*i.e.,* validation accuracy begins to decrease) for five epochs, we stop the training process to prevent increased generalization error.

**Hyper-Parameter Settings.**  We set the causal feature ratio and $\lambda$ as $(r = 0.8, \lambda = 10^{-4}), (r = 0.25, \lambda = 10^{-2}), (r = 0.6, \lambda = 10^2), (r = 0.8, \lambda = 10^{-3})$ for MNIST-75sp, Spurious-Motif, Graph-SST2 and OGBG-Molhiv respectively. For other baselines, we adopt grid search for the best parameters using the validation datasets.

**Model Selection.**  We select each model based on its performance on the corresponding validation dataset. We repeat each experiment at least five times and report the average values and the standard errors in the paper.

# E  UNIMODAL ADJUSTMENT

We follow Cadène et al. (2019) to demonstrate how the shortcut prediction can help to remove model bias. For clarity, we refer to the model parameters except for $\Phi_2$ as the main branch, *i.e.,* except for the $S$-only branch.

Given a house-tree graph as the input graph, we suppose the shortcut prediction $\hat{y}_{\tilde{s}}$ of the tree sub-graph leans towards the house class. Then after reweighting $\sigma(\hat{y}_{\tilde{s}})$ on $\hat{y}_{\tilde{c}}$, the softmax readout on the house class in the joint prediction $\hat{y}$ will be magnified, which results in a smaller loss back-propagated to the main branch and prevents $\hat{y}_{\tilde{c}}$ from inductive bias.

In another situation where a house-wheel graph is given as the input, we similarly suppose the shortcut prediction $\hat{y}_{\tilde{s}}$ of the wheel subgraph leans towards other classes except the house, say, the circle class. Then after reweighting $\sigma(\hat{y}_{\tilde{s}})$ on $\hat{y}_{\tilde{c}}$, the softmax readout on the house class in the joint prediction $\hat{y}$ will be reduced, which results in a larger loss back-propagated to the main branch and encourages the model to learn from these examples.

Furthermore, we offer the causal- and information-theoretical justifications: (1) From the perspective of causal theory (Pearl, 2000; Pearl et al., 2016), the element-wise multiplication enforces the spurious prediction to estimate the pure indirect effect (PIE) of the shortcut features, while the causal prediction captures the natural direct effect (NDE) of the causal patterns (VanderWeele, 2013); (2) From the perspective of information theory (Kullback, 1997), the element-wise multiplication makes the causal prediction reflect the conditional mutual information between the causal patterns and ground-truths, conditioning on the complement patterns.

# F  MORE EXPERIMENTAL RESULTS

## F.1  VISUALIZATION

We provide more visualization cases in Graph-SST2 dataset as shown in Figure 6 and Figure 7. The rationales are highlighted in deep colors.

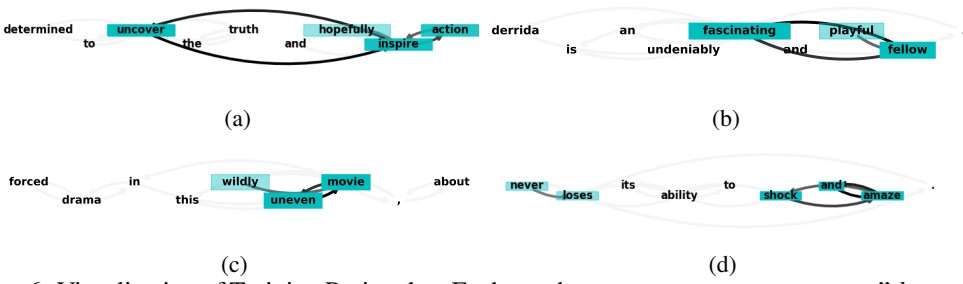

Figure 6: Visualization of Training Rationales. Each graph represents a comment, *e.g., ,"determined to uncover the truth and hopefully inspire action"* in (a).

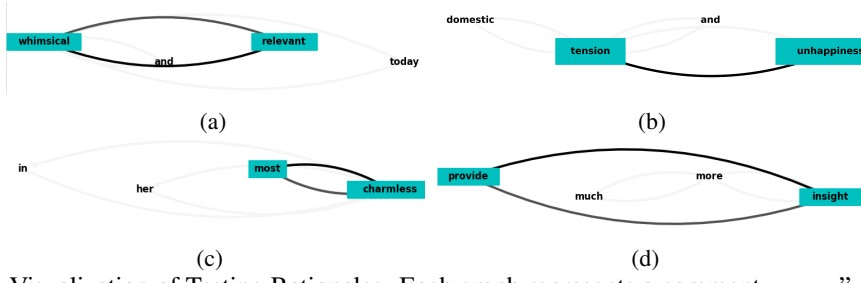

Figure 7: Visualization of Testing Rationales. Each graph represents a comment, *e.g., ,"whimsical and relevant today"* in (a).

Table 4: Confidence of the Spurious Predictions. Uniform is the reference indicates the uniform distributions across the classes.

|  | Spurious-Motif ($b$=0.9) | MNIST-75sp | GraphSST2 | Molhiv |
|---|---|---|---|---|
| Uniform | 1.10 | 2.30 | 0.693 | 0.693 |
| Spurious Predictions | 0.529 | 1.93 | 0.265 | 0.187 |

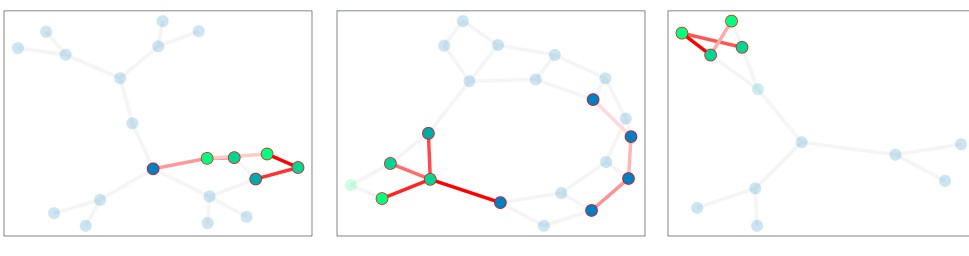

| (a) Cycle-Tree | (b) House-Ladder | (c) Crane-Tree |

Figure 8: Visualization of Training Rationales in Spurious-Motif Dataset. Structures with deeper colors mean higher importance. Nodes of ground truth rationales are colored by green.

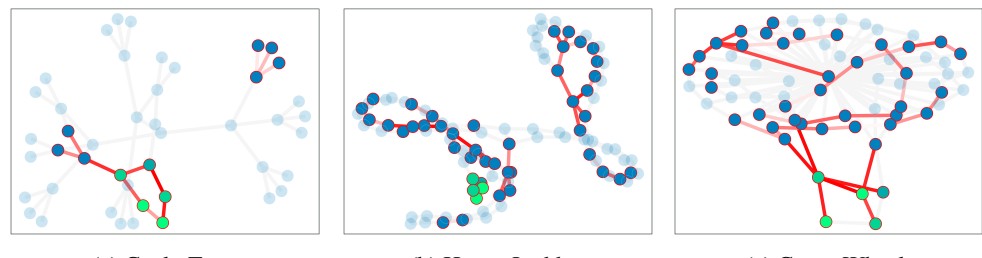

| (a) Cycle-Tree | (b) House-Ladder | (c) Crane-Wheel |

Figure 9: Visualization of Testing Rationales in Spurious-Motif Dataset. Structures with deeper colors mean higher importance. Nodes of ground truth rationales are colored by green.

## F.2 SENSITIVITY ANALYSIS

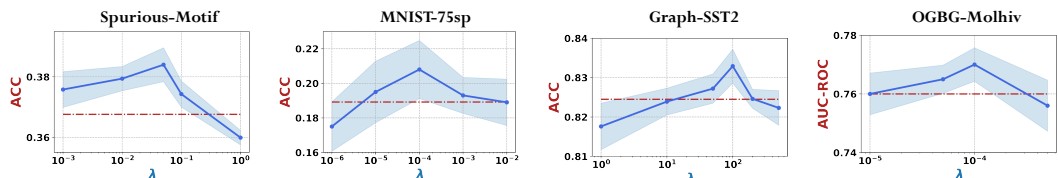

Figure 10: **Sensitivity of Hyper-Parameter $\lambda$.** In each chart, dash line represents the performance of the best baseline in the corresponding dataset, and the area between ACC±std are colored.

We analyze the performance of DIR *w.r.t.* the hyper-parameter $\lambda$. As shown in Figure 10, with $\lambda \to 0$, DIR degrades to optimize the performance in each environment only, without explicitly penalizing the shortcuts' influence on the model predictions. We also see that all testing performances drop sharply if $\lambda$ is too large. Since a large weight on the variance term emphasis on the invariance condition while leading to the overlook on the performance loss, it could fail to exhibit $f_{\tilde{Y}}$ correctly. Notably, such a trade-off in the DIR objective is commonly shared among all the datasets.

## F.3 STUDY OF THE SPURIOUS CLASSIFIERS

Here we provide more observations about the predictions of the learned spurious classifier, which sheds light on the designed model mechanism. We first look into the confidence of predictions and define

$$\nu = \mathbb{E}_{(g,y)\in\mathcal{O},\tilde{s}=g/h_{\tilde{C}}(g)} H\left(\text{Softmax}(\hat{y}_{\tilde{s}})\right) \tag{16}$$

Table 5: Performance of the Spurious Classifiers. $\Delta \downarrow$ indicates the performance gap of the spurious classifiers and the corresponding causal classifiers.

|  | Spurious-Motif ($b$=0.9) | MNIST-75sp | GraphSST2 | Molhiv |
|---|---|---|---|---|
| Spurious Classifiers | $33.43_{\pm 0.22}$ | $17.09_{\pm 0.44}$ | $81.14_{\pm 1.35}$ | $51.13_{\pm 1.29}$ |
| $\Delta \downarrow$ | 6.44 | 3.27 | 2.15 | 25.92 |

where $H$ is the entropy function, and a lower $\nu$ indicates higher confidence. We report the results for the trained spurious classifiers in Table 4. Thus, the results demonstrate the marked tendency of the spurious predictions and validate the design of the $S-$only branch.

However, we show that spurious classifiers are over-confident and potentially overfit to spurious features, which fails to generalize out-of-distribution. In Table 5, we evaluate the spurious classifiers (taking non-causal features as inputs) on the testing sets. We argue that the performance degradation is caused by (i) feature-level problem: it could be theoretically inadequate to infer the label given the non-causal features, and (ii) paradigm-level problem: minimizing the empirical risk only can hardly exhibit stable relations between the features and labels.

### F.4 COMPARISON OF POST-HOC EXPLANATIONS AND INTRINSIC RATIONALES.

Here we aim to compare the explanations generated by GNNExplainer (Ying et al., 2019) and the rationales exhibited by DIR. Specifically, we generate post-hoc explanations from GNNExplainer for Spurious-Motif, where we use the models trained under ERM as the models to explain. We compute the precision of the explanations in Table 6.

Table 6: Explanation/Rationale Accuracy in Spurious-Motif dataset. The results of DIR is consistent with Table 1 and we repeat them here for better view.

|  | Balance | $b$=0.5 | $b$=0.7 | $b$=0.9 |
|---|---|---|---|---|
| GNNExplainer | $0.249_{\pm 0.011}$ | $0.203_{\pm 0.019}$ | $0.167_{\pm 0.039}$ | $0.066_{\pm 0.007}$ |
| DIR | $0.257_{\pm 0.014}$ | $0.255_{\pm 0.016}$ | $0.247_{\pm 0.012}$ | $0.192_{\pm 0.044}$ |

The explanations generated by GNNExplainer reflect the models' inner mechanism, which backs that deep models easily learn from data bias (especially when $b$ is large), being at odds with the true reasoning process that underlies the task. Moreover, even when spurious correlations do not exist, the precisions of rationales generated by DIR still outperform the precisions of the post-hoc explanations, showing the effectiveness of DIR when identifying causal features.

## G  OPEN DISCUSSIONS

Based on this work, we provide open discussions and future directions for the research community, which are inspired by the insightful comments of the ICLR reviewers.

### G.1 EXPRESSIVENESS OF RATIONALE GENERATORS

High expressiveness of the rationale generators could be beneficial for the identification of causal features. Therefore, we have offered additional techniques in our implementation to improve the expressiveness of the graph encoder. Specifically, we incorporate distance encoding measures (Li et al., 2020) like shortest-path distances as the extra node features for better structural representation learning. Also, more powerful graph encoders like RingGNN (Chen et al., 2019) and 3WLGNN (Maron et al., 2019) can be used as the graph encoders to distinguish different substructures better.

## G.2 GENERALIZATION TO UNSEEN SPURIOUS PATTERNS

In our implementation, the memory bank only contains the spurious patterns seen in the training set, while it could possibly fail to unseen spurious patterns. And we provide discussions and solutions to solve this limitation:

- Attribute level perturbation. When the spurious patterns in the testing are different from those in training set only on the attribute level, we can perturb the node/edge attributes of the subgraphs before intervention. And such perturbation is expected to improve the model's robustness during inference.

- External knowledge base. When the spurious patterns also change on the structure level, for example, a star-shaped unseen base graph appears in the testing set of the Spurious-Motif, one potential solution is to resort to prior knowledge. We can enrich the memory bank with possible spurious patterns, *e.g.,* tree (seen) and star (unseen) base graphs. With the external knowledge base, the model can be trained to recognize these possible spurious patterns and be well generalized to the testing dataset.

- Subgraph matching. In a more tricky scenario when the external knowledge base is not available, we can integrate our model with subgraph matching algorithms in the inference. For example, we can extract the training rationales into another bank $\bar{\mathbb{C}}$ and use them to query the testing graphs, *i.e.,* checking if similar patterns exist in the testing graphs. The match results may assist the rationale generator in highlighting the causal features and avoiding unseen spurious features.

## G.3 HIGHER LEVEL INTERPRETABILITY

The interpretability of GNNs in the feature level implicitly demands the separability of a graph into causal and non-causal features. At the same time, we see cases going beyond such assumption (*cf.* Appendix C.1). We believe we could resort to higher level interpretability. For example,

- Interpretability of representations (Wang et al., 2021a; Chan et al., 2021). Instead of highlighting important features for the model decisions, the general goal of representation interpretability answers "What's the information encoded by the $i$-th element of the embedding in the $j$-th layer?".

- Interpretations on top of disentangled variables. Each disentangled latent variable reveals one independent generative factor in the data (Bengio et al., 2013). By generating importance score on these variables, we could possibly obtain more semantically rich interpretations than feature-level interpretability.

Wherein, we believe there are fewer constraints on the separability of features. Thus, the models equipped with higher level interpretability could be applied to a broader range of data-generating assumptions.

