# OpenReview forum: "Discovering Invariant Rationales for Graph Neural Networks"
_ICLR.cc/2022/Conference — ICLR 2022 Poster_

### Official Review · Reviewer_derN · 2021-10-18

**Correctness:** 3
**Technical Novelty And Significance:** 3
**Empirical Novelty And Significance:** 3
**Recommendation:** 8
**Confidence:** 3

**Main Review:**

Strengths:
1. DIR is end2end and novel to me.
2. The DIR principle is intuitive and theoretically justified.
3. The empirical results look good, namely, DIR consistently excels.
4. The observations about how the variance changes during the training course are interesting and provide help for understanding other invariant learning methods.

Weaknesses:
1. What's the motivation behind the element-wise multiplication between causal predictions and spurious predictions? It has not been explained in the paper.
2. With this multiplication and the various sampled spurious subgraphs, it seems that the spurious classifier should output constant vector 1 consistently, so that the causal predictions would not be modulated a lot, right? It would be better to provide more observations in the experiments about the predictions of the learned spurious classifier.

**Summary Of The Paper:**

This paper proposes a novel algorithm DIR that allows the learned GNN model to make predictions based on causal patterns. The whole framework consists of four components where (1) the rationale generator splits the input into causal and spurious parts; (2) the distribution intervener replaces the spurious part; (3) the graph encoder embeds the graph based on the spurious and causal parts, respectively; (4) the shortcut and causal classifiers are applied to the spurious and causal embeddings, respectively. This algorithm is highlighted by its objective which encourages the predictions accurate while invariant to the spurious part. On both synthetic and real-world datasets, DIR outperforms the related baselines consistently. As a novel end2end rationalization method, As a novel end2end rationalization method, the underlying DIR principle is proved to be sufficient and necessary for satisfying the oracle function.

**Summary Of The Review:**

This paper is well written. It proposes a novel end2end algorithm DIR towards causality-based graph representation learning. The algorithm is based on an intuitive principle that is theoretically justified by the authors in this paper. Meanwhile, the extensive experiments also confirm the effectiveness of DIR.

---

> ### Author Response · Authors · 2021-11-18
> **Response to Reviewer derN**
>
> Thank you for your time and insightful suggestions! We are very glad that you have a positive impression on our work. According to your comments, we conduct additional experiments and provide the responses as follows:
>
> **1. The motivation of the element-wise multiplication between causal predictions and spurious predictions.**
>
> We have explained it in Appendix E.1 due to the space limitation (mentioned in the sentence below Eq. (8). Briefly put, the spurious prediction reweights the causal prediction, which dynamically adjusts the loss $l(\hat{y}, y)$ to compensate for biases [1]. Furthermore, we offer the causal- and information-theoretical justifications: (1) From the perspective of causal theory [2,3], the element-wise multiplication enforces the spurious prediction to estimate the pure indirect effect (PIE) of the shortcut features, while the causal prediction captures the natural direct effect (NDE) of the causal patterns [4]; (2) From the perspective of information theory [5], the element-wise multiplication makes the causal prediction reflect the conditional mutual information between the causal patterns and ground-truths, conditioning on the complement patterns.
>
> **2. Observations about the spurious predictions.**
>
> We respectfully argue that the spurious classifiers are not expected to output constant vectors, as the spurious branch is only optimized via the empirical risk (Eq. 9). Instead, the spurious-classifier will capture the bias introduced by non-causal features, which outputs confident predictions with obvious tendencies.
>
> We have followed your suggestions to conduct new experiments \textit{w.r.t.} the spurious classifiers, in which we first define indifference of classifier as
>
> $$\nu:=\mathbb{E}_{(g, y)\in\mathcal{O},\tilde{s}=g/h _{\tilde{c}} (g)}H\left(\text{Softmax}(\hat{y} _{\tilde{s}})\right)$$
>
>
> where $H$ is the entropy function, and a lower $\nu$ indicates higher confidence. And we include Uniform as the constant prediction and report the results for the trained spurious classifiers in all datasets as follows.
>
> | | Spmotif (b=0.9)|MNIST75sp |GraphSST2 | Molhiv
> |:----------------:|:----|:----|:--------|:--------|
> |Suprious Predictions |0.529|1.93|0.265|0.187|
> | Uniform|1.10|2.30|0.693|0.693|
>
> Thus, the results validate the marked prediction tendency of spurious classifiers. Further, we show that spurious classifiers are over-confident and potentially overfitting to spurious features, which fails to generalize out-of-distribution. Specifically, we evaluate the spurious classifiers (when taking non-causal features as inputs) on the testing sets as follows
>
> | | Spmotif (b=0.9)|MNIST75sp |GraphSST2 | Molhiv |
> |:---------:|:----|:--------|:--------|:--------|
> |Spurious Classifiers|33.43 $\pm$ 0.22|17.09 $\pm$ 0.44|81.14 $\pm$ 1.35|51.13 $\pm$ 1.29|
> |$\Delta\downarrow$|6.44|3.27|2.15|25.92|
>
> where $\Delta\downarrow$ indicates the performance gap of the spurious classifiers and the corresponding causal classifiers. We argue that the performance degradation is caused by (i) feature-level problem: it could be theoretically inadequate to infer the label given the non-causal features, and (ii) paradigm-level problem: minimizing the empirical risk only can hardly exhibit stable relations between the features and labels.
>
> We thank the reviewer for this comment, and we have included this part in Appendix F.3 to give a more in-depth understanding of the spurious classifiers.
>
> Reference:
>
> [1] Remi Cadene, Corentin Dancette, Hedi Ben-younes, Matthieu Cord, and Devi Parikh. Rubi: Reducing unimodal biases for visual question answering. NeurIPS,2019.
>
> [2] Judea Pearl. Causality: Models, Reasoning, and Inference. 2000.
>
> [3] Judea Pearl, Madelyn Glymour, and Nicholas P Jewell. Causal inference in statistics: A primer. John Wiley & Sons, 2016.
>
> [4] Tyler J VanderWeele. A three-way decomposition of a total effect into direct, indirect, and interactive effects. Epidemiology (Cambridge, Mass.), 24(2):224, 2013.
>
> [5] Solomon Kullback. Information theory and statistics. Courier Corporation, 1997.

---

> > ### Comment · Reviewer_derN · 2021-11-19
> > **Discussion**
> >
> > Thanks for your reminder. I carefully read E.1 and found that it is very intuitive. With the element-wise multiplication, the minority of spurious patterns tends to produce a significant change for the causal branch, which pushes the model to rely on the causal pattern more heavily. I also noticed the concerns and corner cases mentioned in reviewer1's comments. They are very insightful, from which I learn a lot. Moreover, I appreciate your discussions and believe some interesting questions deserve to be addressed in the future. As for now, I think the presented progress are helpful enough for the community.

---

> > > ### Author Response · Authors · 2021-11-20
> > > **Thank you**
> > >
> > > Thank you! We are very grateful for your approval. And we will continue to refine our work and develop more solid extensions based on these insightful comments.

---

### Official Review · Reviewer_qBQC · 2021-11-03

**Correctness:** 4
**Technical Novelty And Significance:** 3
**Empirical Novelty And Significance:** 4
**Recommendation:** 8
**Confidence:** 4

**Main Review:**

Strengths:
The tackled problem of identifying the environment-invariant causal patterns for improving the interpretability of graph neural networks is interesting and important for many graph-structured data representation tasks.

The proposed DIR model is technically sound with the construction of interventional distributions and discover the salient features from different environments. In addition, the theoretical discussion is provided to analyze the proposed DIR method.

In the evaluation section, the effectiveness of the new DIR model is analyzed from different aspects: model generalization ability, performance with different bias degrees, as well as the intrinsic interpretability. Furthermore, the identified rationale has been visualized under interventional distributions.

Weaknesses:

It would be better to summarize the key notations used throughout this paper in a table, for easy reference when reviewer go through the solution details.

The detailed configurations of the compared baselines could be described in the evaluation so as to provide a more comprehensive performance comparison.

**Summary Of The Paper:**

This paper proposes to address the limitation of current interpretable graph neural networks over out-of-distribution data. In this work, the intrinsically interpretable GNN is investigated by identifying the invariant rationales corresponding to environment-invariant causal patterns. The proposed model is evaluated on the graph classification task to demonstrate its effectiveness as compared to state-of-the-arts.

**Summary Of The Review:**

The studied problem is important and beneficial for many graph-learning tasks. The proposed method is rationale and technically sound. More details about the experimental settings could be clarified in the evaluation section.

---

> ### Author Response · Authors · 2021-11-18
> **Response to Reviewer qBQC**
>
> We are grateful for your approval. Please find the responses to the specific comments.
>
> **1. Summarize the key notations.**
>
> Thanks. We followed your suggestion to provide a table of notations in Appendix A due to the current space limitation of the main paper.
>
> **2. Detailed configurations of the compared baselines.**
>
> We have stated the detailed experimental setting in Appendix D, including the backbone models, training details, and model selection methods, which are the same for the baseline methods and DIR. We also include the hyper-parameter settings for all methods in our code for reproducibility.

---

### Official Review · Reviewer_F9KM · 2021-11-08

**Correctness:** 3
**Technical Novelty And Significance:** 2
**Empirical Novelty And Significance:** 2
**Recommendation:** 6
**Confidence:** 4

**Main Review:**

*This is an emergency review*

Update Nov 20, 2021:

Recommendation: Weak Accept

Increasing my score to 6 after some discussions with the authors.

***************************************
Initial Recommendation: Weak Reject

I have tried to reason my recommendation below:

Strengths:
1. The idea to use a causal graph and the accompanying theory to explain the label associated with the graph is interesting.
2. The paper is well written and easy to understand.

Weaknesses
1. Via the top-K approach, it appears like the authors make an implicit assumption about what fraction of edges is important in the graph (a fixed fraction, can't be lower or higher) - in my opinion, this certainly limits the wide applicability of the method. Alternatively, in GNN Explainer [1], they allow up to K edges to explain the label.
2. The applicability of the method is limited by the graph encoder used (here GNN). 1-WL GNN's are known to be unable to predict links or identify and distinguish certain classes of subgraphs [2][3][4] i.e. GNN's are not well suited for Eq. 5
3. The work appears to miss relevant baselines like GNN Explainer[1] / CF-GNN Explainer [5]. Moreover the authors ideally should compare with baselines, which enrich GNN's with structural features [6][7][8] (As they are explainable to a certain extent as well).

Minor:
1. The separability of a graph into two subgraphs the causal and non causal might not always be possible? (would this need an encoder which is able to accurately capture the discrete topology over graphs of all orders and sizes - if not I can just make a house into a clique and the label would be incorrectly assigned as 1 because the edges required for the house are present) Please elaborate on this.
2. The set  {s} employed in the test are limited to the ones seen in train.
3. Moreover, consider a graph with say 20 nodes, and the case where the causal part for instance consists of a house motif and a tree base and the label assigned is 1 or 0 based on the output of House Motif XOR Tree Base - would this be captured by the proposed method (appears like it wont)?



References:

1.Ying, Rex, et al. "Gnn explainer: A tool for post-hoc explanation of graph neural networks." arXiv preprint arXiv:1903.03894 (2019).

2.Srinivasan, Balasubramaniam, and Bruno Ribeiro. "On the equivalence between positional node embeddings and structural graph representations." arXiv preprint arXiv:1910.00452 (2019).

3.Dwivedi, Vijay Prakash, et al. "Benchmarking graph neural networks." arXiv preprint arXiv:2003.00982 (2020).

4.Chen, Zhengdao, et al. "Can graph neural networks count substructures?." arXiv preprint arXiv:2002.04025 (2020).

5.Lucic, Ana, et al. "CF-GNNExplainer: Counterfactual Explanations for Graph Neural Networks." arXiv preprint arXiv:2102.03322 (2021).

6.Bouritsas, Giorgos, et al. "Improving graph neural network expressivity via subgraph isomorphism counting." arXiv preprint arXiv:2006.09252 (2020).

7.Bodnar, Cristian, et al. "Weisfeiler and lehman go topological: Message passing simplicial networks." arXiv preprint arXiv:2103.03212 (2021).

8.Bodnar, Cristian, et al. "Weisfeiler and lehman go cellular: Cw networks." arXiv preprint arXiv:2106.12575 (2021).

**Summary Of The Paper:**

The authors propose to improve generalization ability of GNN's for the task of graph classification via a causal theoretic analysis - under the assumption that only a subgraph of the graph is responsible for its label (and everything else are just spurious correlations). To this effect, they use a rationale generator and distribution intervener to test their model on a few synthetic datasets and a real world dataset.

**Summary Of The Review:**

The paper is well written and the proposed solution is simple but there are still concerns about the wide applicability of the method, and also lacks comparisons with very related baselines.

---

> ### Author Response · Authors · 2021-11-18
> **Response to Reviewer F9KM**
>
> We appreciate your comments! To address your concerns, below we prudently justify the applicability of our proposed method, clarify the misunderstandings, and conduct more experiments.
>
> **1. The implicit assumption about the fraction of causal edges.**
>
> Thanks for your valuable comments. This concern may be raised by the misunderstanding of Eq. (6). Note the hard selection of causal edges does not eliminate the soft masks on them. And the soft mask value on an edge directly controls the strength of the message-passing between the end nodes (it can be very low, which means the edge barely passes any message). In other words, a fine-grained edge fraction is available within the hard selection ratio.
>
> On another hand, following previous methods [1,2,3,4,5], $r$ can be regarded as an additional hyper-parameter in our framework. As stated in Appendix D, we adopted grid search for the best ratios in the validation datasets. Thus, it does not require pre-knowledge to determine $r$, and can be widely applied.
>
> Moreover, inspired by your concern, we tried another implementation that allows various causal ratios for graphs, where the generator outputs the vector $\mathbf{M}' \in \mathbb{R}^{|\mathcal{E}|\times 2}$. Each row of $\mathbf{M}' $ conforms to a gumbel-softmax distribution and indicates the probability for the edge to be included in the causal or spurious part. We assign each edge to the part with maximum probability, making the causal ratio automatically adjust for each graph. In training, we additionally include the $L_1$ penalty on the rationale size to avoid a naive solution (i.e., assign all edges to the causal part).
>
> However, we found the model trained under the dynamic causal ratio consistently underperforms the previous one. The model only achieves an accuracy of 34% in Spurious-Motif (b=0.9), while the original model gains 40%.
>
> This empirical result is consistent with the conclusion of the two-stage training dynamics proposed in Section 3.3, where we found an adaption-fitting pattern in the learning process of DIR. Specifically, we found the rationales are exhibited early in the training stage; then, the model mainly learns a good prediction function from the rationales to the labels. We believe that the dynamic ratio disrupts the balance of such two-stage learning since the ratio constantly changes during training, which results in unstable and poor outcomes.
>
> **2. The applicability of the method is limited by the graph encoder.**
>
> Thanks for your insightful suggestions. In our experiments, we used k-GNNs [7] as the encoder for MNIST-75sp, which are strictly more powerful than1-WL GNNs [7,8].
>
> Moreover, following your suggestions, we use additional techniques and potential solutions to improve the expressiveness of the graph encoder and include this part in Appendix E.2.
>
> We use k-GNNs [7] as the graph encoders in all datasets and provide distance encoding (DE) measures [9] like shortest-path distances as the extra node features (optional) for better structural representation learning. The results are shown as follows, where we find some improvements on Spurious-Motif and GraphSST2. We also update the code accordingly for reference.
>
> | |Spurious-Motif (Blance)| Spurious-Motif ($b$=0.5)| Spurious-Motif ($b$=0.7)| Spurious-Motif ($b$=0.9)|MNIST-75sp |GraphSST2 |Molhiv|
> |:---------:|:--------|:--------|:--------|:--------|:--------|:--------|:--------|
> |DIR|47.03 $\pm$ 2.46|45.50 $\pm$ 2.15|43.36 $\pm$ 1.64|39.87 $\pm$ 0.56|20.36 $\pm$ 1.78|83.29 $\pm$ 0.53|77.05 $\pm$ 0.57|
> |DIR+DE|47.31 $\pm$ 2.01|46.45 $\pm$ 1.52|43.25 $\pm$ 0.98|40.11 $\pm$ 1.03|18.65 $\pm$ 1.33|83.57 $\pm$ 0.21|76.83 $\pm$ 0.88|
>
> Also, more powerful graph encoders like RingGNN [10] and 3WLGNN [11] can be used as graph encoders in our future work.

---

> > ### Author Response · Authors · 2021-11-18
> > **Response to Reviewer F9KM (cont.)**
> >
> > **3. Missing relevant baselines.**
> >
> > Thanks for your suggestions. However, the post-hoc explainability is different from the intrinsic interpretability, hence the post-hoc explainer (e.g., GNNExplainer, CF-GNNExplainer) is out of the scope of our work, since they neither offer interpretations/rationales along with the prediction nor directly improve the generalization ability of GNN models. But we find it might be a good idea to compare the post-hoc explanations and intrinsic rationales.
> >
> > Specifically, we generate post-hoc explanations from GNNExplainer for Spurious-Motif, where we use the models trained under ERM as the models to explain. We compute the precision of the explanations as follows.
> >
> > |Model|Balance| $b$ = 0.5| $b$ = 0.7| $b$ = 0.9|
> > |:----------------:|:--------------|:-------------|:------------|:------------|
> > |GNNExplainer |0.249 $\pm$ 0.011| 0.203 $\pm$ 0.019 |0.167 $\pm$ 0.039| 0.066 $\pm$ 0.007|
> > |DIR |0.257 $\pm$ 0.014| 0.255 $\pm$ 0.016| 0.247 $\pm$ 0.012 |0.192 $\pm$ 0.044|
> >
> > As the explanations generated by GNNExplainer reflect the models' inner mechanism, the results support that deep models easily learn from data bias (when $b$ is large), being at odds with the true reasoning process that underlies the task. Moreover, even when spurious correlations do not exist, the precisions of rationales generated by DIR still outperform the precisions of the post-hoc explanations, showing the effectiveness of DIR when identifying causal features. We have included the results and analysis in Appendix F.
> > For CF-GNNExplainer, it seems inapplicable under our setting as the CF examples can't be evaluated by precision.
> >
> > Also, thank you for bringing the other baselines to us. We run additional experiments on GSN [6] for five times, where the models are under the same parameter sizes as those in our paper, and report the results as follows
> >
> > | |Spurious-Motif (Blance)| Spurious-Motif ($b$=0.5)| Spurious-Motif ($b$=0.7)| Spurious-Motif ($b$=0.9)|MNIST-75sp |GraphSST2 |Molhiv|
> > |:---------:|:--------|:--------|:--------|:--------|:--------|:--------|:--------|
> > |GSN [6]|43.18 $\pm$ 5.65|34.67 $\pm$ 1.21|34.03 $\pm$ 1.69|32.60 $\pm$ 1.75|19.03$\pm$ 2.39|82.54 $\pm$ 1.16|74.53 $\pm$ 1.90|
> > |DIR|47.03 $\pm$ 2.46|45.50 $\pm$ 2.15|43.36 $\pm$ 1.64|39.87 $\pm$ 0.56|20.36 $\pm$ 1.78|83.29 $\pm$ 0.53|77.05 $\pm$ 0.57|
> >
> > Specifically, in MNIST-75sp, although structure features are shown to be helpful in mitigating feature distribution shift, DIR still performs better than GSN. And we have updated our paper accordingly.
> >
> >
> > **4. Separability of a graph.**
> >
> > We have discussed this problem in Appendix C.1 towards the plausibility and limitations of our assumptions. We moved the discussions here for your convenience.
> >
> > We argue that the assumption (c.f. Assumption 1) is commonly satisfied. For example, for sentences labeled by sentiment, $C$ can represent the positive/negative words that cause the sentiment, while $S$ includes the prepositions and linking words. For molecule graphs labeled by specific properties, $C$ and $S$ can represent the functional groups and carbon structures, respectively. Note that IR assumption enables and calls the introduction of interpretability, highlighting salient features and exhibiting human accessible checks. More importantly, it guarantees the model performance under possible feature reduction, i. e., $C\subset G$.
> >
> > Surely there are cases going beyond the IR Assumption. For example, $G$ could be a generic function of $S$ and $C$, instead of a simple joint. Thus, individual $S$ and $C$ may be intractable individually in the feature level. In such a scenario, the separation of the features could degrade model performance due to information loss.
> >
> > **5. The set ${s}$ employed in the testing is limited to the ones seen in the training.**
> >
> > No. We don't use $s$ in the testing. As stated at the bottom of page 5, in the inference phase, we yield $\tilde{c}$ and $\hat{y}_{\tilde{c}}$ as the causal rationale and the causal prediction of a testing graph $g$, which exclude the influence of the non-causal part $\tilde{s}$.
> >
> > **6. The case where graphs are labeled as House Motif XOR Tree Base.**
> >
> > Good catch. In your case, the causal part $C$ includes the full graph, and the non-causal part $S$ is an empty set. Thus, there is no spurious correlation between $S$ and $Y$, and the distribution shift can only happen inside $C$.
> > And this is out of the scope of our data-generating process (Figure 2). Although this is not the focus of our work, we believe it will be great to explore more complicated data-generating processes in future work.

---

> > > ### Author Response · Authors · 2021-11-18
> > > **Reference**
> > >
> > >
> > > [1] Hongyang Gao, Shuiwang Ji. Graph U-Nets. ICML 2019.
> > >
> > > [2] Cangea et al. Towards Sparse Hierarchical Graph Classifiers. NIPS 2018 workshop.
> > >
> > > [3] Lee et al. Self-Attention Graph Pooling. ICML 2019.
> > >
> > > [4] Knyazev et al. Understanding Attention and Generalization in Graph Neural Networks. NeurIPS 2019.
> > >
> > > [5] Ranjan et al. ASAP: Adaptive Structure Aware Pooling for Learning Hierarchical Graph Representations. AAAI 2020.
> > >
> > > [6] Bouritsas, Giorgos, et al. "Improving graph neural network expressivity via subgraph isomorphism counting." arXiv preprint arXiv:2006.09252 (2020).
> > >
> > > [7] Morris et al. Weisfeiler and Leman Go Neural: Higher-order Graph Neural Networks. AAAI 2019.
> > >
> > > [8] Dwivedi et al. Benchmarking graph neural networks. arXiv:2003.00982.
> > >
> > > [9] Li et al. Distance Encoding: Design Provably More Powerful Neural Networks for Graph Representation Learning. NeurIPS 2020.
> > >
> > > [10] Chen et al. On the equivalence between graph isomorphism testing and function approximation with gnns. NeurIPS 2019.
> > >
> > > [11] Maron et al. Provably powerful graph networks. NeurIPS 2019.

---

> > > > ### Comment · Reviewer_F9KM · 2021-11-19
> > > > **Reply to the authors**
> > > >
> > > > Thank you very much for the reply.
> > > >
> > > > I still have some questions, this is with regard to 4 and 5 in your responses.
> > > >
> > > > With respect to 5 - I believe there is some misunderstanding between my question and the answer. What I had meant is that, the set of interventions (based on distribution intervener, Pg 5) is based on the spurious $s$ seen before ("all the instances computed previously into a memory bank"). From my understanding, the procedure is not going to recognize spurious $s$  in test if they are unseen in train, and thus can yield an incorrect $c$
> > > >
> > > > With respect 4 - I am sorry, I don't understand. Can you please be more precise - a specific example would be very helpful (the house to clique, for example).

---

> > > > > ### Author Response · Authors · 2021-11-20
> > > > > **Further Response**
> > > > >
> > > > > Thank you for letting us know! Please see the following discussions for the answers.
> > > > >
> > > > > **(cont.) 4. Separability of a graph.**
> > > > >
> > > > > Thanks! Yes, the generator is required to learn the discrete topology of graphs, so that it can distinguish the "house in the clique" and the "single house motif". More precisely, for a well-trained generator, its message-passing scheme of the "house in the clique" is different from that of the "single house motif", since the "house in the clique" is denser and featured differently. As such, the generator is able to separate a graph into the causal and non-causal parts.
> > > > >
> > > > > About the statement we made in the last response ("Surely there are cases going beyond the IR Assumption..."), we meant to answer your first question ("The separability of a graph into the causal and non-causal subgraphs might not always be possible?"). And the answer is: **yes, it is not always possible**. There do exist cases where the graph can not be separated, even with a strong generator. Here we construct a toy example for better clarification.
> > > > >
> > > > > For example, assume each graph has multiple motifs (house, cycle, crane) with only one type and is labeled by the motif type. Thus, the causal feature $C$ will be the motifs. Let the spurious feature $S$ be "the way we connect the motifs". For example, we can place the house motifs in a queue sequence (say, from left to right) and connect the adjacent motifs, thus forming the graph in a "line" shape. Or we can place the houses in a cycle order and connect them into a ring. We further let such graph structures be strongly correlated with the motif type. Clearly, we can not separate the graphs. For example, if we separate the cycle-shaped houses into two lines, the spurious pattern could be broken while the part of the causal feature would be lost. In other words, $S$ and $C$ are dependent variables (we referred to such cases as "$G$ is a generic function of $S$ and $C$" in our last response). Thus, they can't be extracted and modeled separately, even with a generator that can capture the discrete topology well.
> > > > >
> > > > >
> > > > > **(cont.) 5. Generalization to unseen spurious patterns**
> > > > >
> > > > > Thank you for re-explaining your question! In this sense, yes, the memory bank only contains the spurious patterns seen in the training set, while our DIR framework possibly fails to unseen spurious patterns. To solve this limitation, We list several solutions as follows.
> > > > >
> > > > > 1. **By attribute level perturbation.** When the spurious patterns in the testing are different from those in training set only on the attribute level (node attributes or edge attributes), we can perturb the node/edge attributes of the subgraphs before intervention. Such perturbation can improve the model's robustness when inference.
> > > > >
> > > > > 2. **By an external knowledge base.** When the spurious patterns also change on the structure level, for example, a star-shaped unseen base graph appears in the testing set, one potential solution is to resort to prior knowledge. We can enrich the memory bank with potential spurious patterns, e.g., tree (seen) and star (unseen) base graphs. With the external knowledge base, the model can be trained to recognize these possible spurious patterns and be well generalized to the testing dataset.
> > > > >
> > > > > 3. **By subgraph matching.** In a more tricky scenario when the external knowledge base is not available, we can integrate our model with subgraph matching algorithms in the inference. For example, we can extract the training rationales into another bank and use them to query the testing graphs (see if similar patterns exist in the testing graphs). The match results may assist the rationale generator in highlighting the causal features and avoiding unseen spurious features.
> > > > >
> > > > > Feel free to let us know if you have more questions!

---

> > > > > > ### Comment · Reviewer_F9KM · 2021-11-20
> > > > > > **Reply to the authors**
> > > > > >
> > > > > > Thank you very much for the reply.
> > > > > > At this moment, I do not have any more questions (which would need you to update the pdf) - but will discuss further with the AC and other reviewers as well.

---

### Author Response · Authors · 2021-11-18
**General response**

We sincerely appreciate all reviewers' time and efforts in reviewing our paper. We are glad that most reviewers have a positive impression of our work. We would like to thank all reviewers for providing many insightful and valuable suggestions. Here is a summary of our updates:
1. Clarification: We justify the technical detail (derN) and the applicability underlying our assumption (F9KM), and add key notations for better clarity (qBQC).
2. More Experiments: We offer empirical observation for the spurious predictions (derN) to better understand the model component. Also, we improve the expressiveness of the graph encoders, provide a comparison for post-hoc explanations and intrinsic rationals, and add the relevant baseline (F9KM).

We've highlighted the updates in the revision. We hope our responses can clarify all your confusion and alleviate all concerns. We thank all reviewers' time again. Looking forward to your reply!

---

### Author Response · Authors · 2021-11-23
**Paper Updating**

We again appreciate all reviewers’ approval and replies. We further update our paper by adding an **Open Discussions** section in Appendix G, which presents some promising future directions to further enhance our DIR framework. We sincerely hope this can make our work more sound and offer more inspiration for the community.

---

### Decision · Program_Chairs · 2022-01-20

**Decision:**

Accept (Poster)

**Comment:**

The work proposes a method to learn graph representations based on subgraphs that are invariant to spurious subgraphs. The reviewers found the paper easy to read and the theory interesting, well explained and justified. The reviewers seem happy with the existing and new experiments that came during the rebuttal phase. I too found the paper interesting and mostly well-written.

Besides the corrections done during the rebuttal, in further discussion with the authors, I raised a concern that the work must make additional assumptions about the support of the induced subgraph distributions that were not clearly stated in the paper: The work makes the assumption that there is enough training data such that all spurious induced subgraph patterns $S$ that are smaller than the truly correlated induced subgraph $C$ can be identified as spurious. The authors promised to make this into a clearly demarcated assumption since it a key requirement for the method to work.